# Sterol preservation in hypersaline microbial mats

Yan Shen[1], Volker Thiel[1], Pablo Suarez-Gonzalez[2], Sebastiaan W. Rampen[1], Joachim Reitner[1,3]

[1]Department of Geobiology, Geoscience Centre, Georg-August-Universität Göttingen, Göttingen, Germany
[2]Área de Geología, Universidad Rey Juan Carlos, Madrid, Spain
[3]'Origin of Life' Group, Göttingen Academy of Sciences and Humanities, Göttingen, Germany

*Correspondence to*: Yan Shen (yshen@gwdg.de)

**Abstract.** Microbial mats are self-sustaining benthic ecosystems composed of highly diverse microbial communities. It has been proposed that microbial mats were widespread in Proterozoic marine environments, prior to the emergence of bioturbating organisms at the Precambrian-Cambrian transition. One characteristic feature of Precambrian biomarker records

is that steranes are typically absent or occur in very low concentrations. This has been explained by low eukaryotic source inputs, or degradation of primary produced sterols in benthic microbial mats ("mat-seal effect"). To better understand the preservational pathways of sterols in microbial mats we analysed freely extractable and carbonate-bound lipid fractions as well as decalcified extraction residues in different layers of a recent calcifying mat (~1500 years) from the hypersaline Lake 2 on the island of Kiritimati, Central Pacific. A variety of $C_{27}$-$C_{29}$ sterols and distinctive $C_{31}$ 4α-methylsterols (4α-

methylgorgosterol and 4α-methylgorgostanol, biomarkers for dinoflagellates) were detected in freely extractable and carbonate-bound lipid pools. These sterols most likely originated from organisms living in the water column and the upper mat layers. This autochthonous biomass experienced progressive microbial transformation and degradation in the microbial mat, as reflected by a significant drop in total sterol concentrations, up to 98 %, in the deeper layers, and a concomitant decrease in total organic carbon. Carbonate-bound sterols were generally low in abundance as compared to the freely

extractable portion, suggesting that incorporation into the mineral matrix does not play a major role for the preservation of eukaryotic sterols in this mat. Likewise, pyrolysis of extraction residues suggested that sequestration of steroid carbon skeletons into insoluble organic matter was low as compared to hopanoids. Taken together, our findings argue for a major 'mat-seal effect' affecting the distribution and preservation of steroids in the mat studied. This result markedly differs from recent findings made for another microbial mat growing in the near-by hypersaline Lake 22 on the same island, where sterols

showed no systematic decrease with depth. The observed discrepancies in the taphonomic pathways of sterols in microbial mats from Kiritimati may be linked to multiple biotic and abiotic factors including salinity and periods of subaerial exposure, implying that caution has to be exercised in the interpretation of sterol distributions in modern and ancient microbial mat settings.

# 1 Introduction

Sterols are commonly used as biological markers for specific classes of organisms (Atwood et al., 2014; Brocks and Summons, 2004; Rampen et al., 2009; Volkman, 1986; Volkman, 2005). Sterols have been found in many different types of depositional environments such as soils (van Bergen et al., 1997; Birk et al., 2012; Otto and Simpson, 2005), recent lacustrine and marine sediments (Brassell and Eglinton, 1983; Gaskell and Eglinton, 1976; Robinson et al., 1984; Volkman, 1986), as well as microbial mats from meso- to hypersaline conditions (Grimalt et al., 1992; Scherf and Rullkötter, 2009). Further, the hydrocarbon skeleton of sterols is relatively stable, and thus significant amounts can be preserved in the geological record (Brocks et al., 2017; Mattern et al., 1970).

Microbial mats are vertically laminated organo-sedimentary structures, which are primarily self-sustaining ecosystems (Des Marais, 2003), ranging in thickness from millimeters to decimeters. The mineralized fossil product of microbial mats are microbialites, which have a long geological history of over 3 billion years, indicating that microbial mats probably represented the earliest complex ecosystems on Earth (Reitner and Thiel, 2011). Microbial mats typically consist of many different functional groups of microorganisms which control the organic matter (OM) turnover in the microbial mat. Major groups include cyanobacteria, colorless sulfur bacteria, purple sulfur bacteria and sulfate-reducing bacteria, but also eukaryotic organisms (Schneider et al., 2013; van Gemerden, 1993). A large proportion of the OM consists of extracellular polymeric substances (EPS), secreted by the microorganisms, which are crucial for the support and the development of the microbial mat (Decho, 2011; Reitner and Thiel, 2011; Wingender et al., 1999). EPS are rich in acidic groups that bind cations such as $Ca^{2+}$, thereby inducing a strong inhibitory effect on the precipitation of common minerals formed within microbial mats, such as $CaCO_3$ (Arp et al., 1999; Dupraz et al., 2009; Ionescu et al., 2015). Consequently, carbonate precipitation often occurs in deeper and older mat layers in which decomposing EPS gradually releases previously-bound $Ca^{2+}$, thus facilitating carbonate supersaturation (Arp et al., 1999; Dupraz et al., 2009; Ionescu et al., 2015). Previous studies indicate that early sequestration into a mineral matrix may promote the preservation of organic compounds (Summons et al., 2013; Smrzka et al., 2017; Thiel et al., 1999). Hence, microbial mats possibly provide an enhanced chance for OM to survive in the geosphere if carbonate or other mineral precipitation occurs therein.

Microbial mats have been proposed to be a predominant life form in the Proterozoic marine environments. In contrast, the Phanerozoic is characterized by prosperity of biota including fauna and flora, and a low abundance of benthic microbial mats (Grotzinger and Knoll, 1999; Riding, 2011; Walter, 1976). One of the characteristic features of the Precambrian biomarker records is that eukaryotic steranes are typically absent or occur in very low concentrations. This may be explained by a limited ecological distribution of eukaryotic algae and thus minor contributions of sterols to sedimentary OM (Anbar and Knoll, 2002; Blumenberg et al., 2012; Brocks et al., 2017; Knoll et al., 2007), and/or by a thermal degradation of sterols during catagenesis, as observed for the 1640 Ma Barney Creek Formation and 1430 Ma Velkerri Formation, Northern Australia (Dutkiewicz et al., 2003; Summons et al., 1988). An alternative explanation would be that eukaryotic lipids have been subject to a preservation bias due to the ubiquity of benthic microbial mats. It has been hypothesized that these mats

would have formed a significant mechanical and chemical barrier against the preservation of eukaryotic lipids sourced from water column and upper mat layers, a phenomenon termed as "mat-seal effect" (Pawlowska et al., 2013). Selective preservation induced by the mat-seal effect would also impart a bias in favour of lipids derived from heterotrophic microorganisms living in the deeper mat layers, and cause a suppression of the primary ecological signal. This is different from the situation in the Phanerozoic, where OM from planktonic primary producers (including algae and bacteria) is more rapidly transferred to the sediment through sinking aggregates (such as crustacean faecal pellets), without being reworked in benthic microbial mats (Close et al., 2011; Fowler and Knauer, 1986; Logan et al., 1995).

The Kiritimati atoll (Kiribati Republic, Central Pacific, Fig. 1) is an ideal study site for investigating the taphonomy of sterols in microbial mats. The island is covered by c. 500 brackish to hypersaline lakes, most of which are populated by thick and highly developed benthic mats that are clearly laminated and show ongoing mineral precipitation, i.e. microbialite formation (Arp et al., 2012; Trichet et al., 2001; Valencia, 1977). Therefore, Kiritimati enables studies on the behaviour of sterols within various types of microbial mats thriving under different environmental conditions and showing different degrees of mineralization.

A recent study conducted on a microbial mat from Lake 22 on Kiritimati demonstrated that a range of sterols were abundantly present in all parts of that mat (Shen et al., 2018a). The lack of any systematic decrease with depth suggested that the sterols in that particular mat had not been impacted by a major mat-seal effect. On the other hand, an earlier study on insoluble OM obtained from a microbial mat from a different lake of the same island (Lake 2, located about 10 km south of Lake 22) reported an increasing trend of hopane/sterane ratios with depth (Blumenberg et al., 2015). In conjunction with other findings, this was considered indicative of a "*suppression of biosignatures derived from the upper mat layers*" and thus, a mat-seal effect (Blumenberg et al., 2015). Since that work had a different focus and did not report detailed sterol data, it is not directly comparable with the results on the Lake 22 mat reported by Shen et al. (2018a). Therefore we revisited the microbial mat from Lake 2 and performed a detailed analysis of sterol compounds, investigating both freely extractable as well as carbonate-bound lipid fractions, and also decalcified extraction residues (here refers to kerogen fraction). Our study was aimed at further examining general trends in the preservation of sterols in hypersaline microbial mat systems by comparing the results from different settings within the same geological and geographical context (i.e. Lakes 2 and 22).

## 2 Materials and methods

### 2.1 Location and samples

The atoll of Kiritimati (Republic of Kiribati) is located in the central part of the Pacific Ocean, close to the Equator (Fig. 1). Its surface displays a complex reticular pattern encompassing c. 500 lakes with salinities that range from brackish to hypersaline. Most of the lakes harbour thick microbial mats that show ongoing mineralization processes (Figs. 1, 2) and generally occur on top of older, more developed microbialites (i.e. already fossilized microbial mats; Arp et al., 2012;

Ionescu et al., 2015; Trichet et al., 2001; Valencia, 1977). Vegetation around the lake areas comprises the mangrove *Rhizophoramucronata*, the parasitic climber *Cassytha filiformis*, the grass *Lepturus repens*, and the ironwood *Pemphis acidula* (Fig. 2e; Saenger et al., 2006). The climate of Kiritimati is broadly controlled by the El Niño-Southern Oscillation (ENSO) atmospheric phenomenon. During El Niño wet events, heavy rains occur, decreasing lake salinities; whereas reduced precipitation during La Niña dry events triggers higher evaporation and increasing lake salinities (Arp et al., 2012; Saenger et al., 2006; Trichet et al., 2001). Materials studied in this work were sampled from Lake 2 (Fig. 1), whose salinity was 97 ‰ in 2002 and 125 ‰ in 2011 (own data, unpublished). This high and variable salinity causes low metazoan diversity within Lake 2. Faunal elements include abundant *Tilapia* fish (Fig. 2d) and *Artemia* brine shrimp as well as few land crabs, and unicellular miliolid foraminifera (Saenger et al., 2006; Shen et al., 2018a). Events of mass mortality of fish have been observed in some of the lakes, including Lake 2 (Fig. 2d), which may be linked to extreme hypersaline conditions probably due to heavy evaporation during La Niña dry periods. More detailed information about the environmental setting of Kiritimati can be found elsewhere (Arp et al., 2012; Saenger et al., 2006; Shen et al., 2018a; Trichet et al., 2001).

In this work, a microbial mat from the hypersaline Lake 2, previously studied by Blumenberg et al. (2015), was analysed for steroids, hopanoids and fatty acids (Fig. 1). This mat is 10 cm thick and was sampled from the centre of the lake (water depth c. 4 m) during a field campaign in March 2011 (Figs. 1, 2). Samples were stored at -20°C until laboratory preparation. Based on the macroscopic appearance, Blumenberg et al. (2015) divided the mat in five layers, the topmost layer corresponding to the photosynthetically active mat, and layers 2-5 representing ancient mat generations being degraded by recent anaerobic microorganisms (Figs. 1, 2). For this study, we used the same layer division as Blumenberg et al. (2015). However, a thin but distinctive mineral crust occurring just below layer 2 (Fig. 2c) has not been analysed in the previous study and is additionally included here (corresponding to our layer 3, Figs. 1, 2). Therefore, six layers in total were analysed in this work, each one c. 1-2 cm thick (except layer 3 ~0.15 cm).

## 2.2 Bulk analysis

Homogenized (mortar) aliquots of the freeze-dried samples (both original mat layers and extraction residues) were subjected to C/N/S analysis, using a Hekatech EA 3000 CNS analyzer and LECO RC 612 multiphase carbon analyser as described elsewhere (Shen et al., 2018a).

## 2.3 Extraction and derivatization

Aliquots of the freeze-dried samples (5-20 g) were homogenized and extracted using 4×50 ml portions of dichloromethane/methanol (3:1; V/V) (10 min ultrasonication) to obtain the freely extractable lipids. The remaining extraction residues were decalcified using 37 % HCl (dropwise until $CO_2$ development ceased), and again extracted as described above to yield the carbonate-bound lipids. The remaining extraction residues (after decalcification) were freeze dried for the analysis of bulk $C_{org}$ and pyrolysis.

To make alcohols (including sterols and hopanols) GC-amenable, aliquots of the lipid extracts (both freely extractable and carbonate-bound lipid fractions) were silylated using BSTFA (N,O-bis(trimethylsilyl)trifluoroacetamide) containing 5 % (V/V) trimethylchlorosilane (TMCS) as a catalyser (70°C, 60 min). The resulting trimethylsilyl (TMS-) derivatives were dried under gentle $N_2$ flow, re-dissolved in *n*-hexane, and analysed by gas chromatography-mass spectrometry (GC-MS).

To make hopanoids and fatty acids GC-amenable, a mixture of TMCS/MeOH (1:9, V:V; Poerschmann and Carlson 2006) was added to all aliquots of lipid extracts and samples were heated at 80°C for 60 min. The resulting fatty acid methyl esters were extracted from the reaction mixture by vigorous shaking with 3×1 ml *n*-hexane. The extracts were combined and evaporated to near-dryness under a gentle stream of $N_2$, re-dissolved in *n*-hexane, and analysed by gas chromatography-mass spectrometry (GC-MS).

**2.4 GC-MS**

GC-MS analyses were carried out using a Thermo Fisher Trace 1310 GC coupled to a Thermo Fisher Quantum XLS Ultra MS as described elsewhere (Shen et al., 2018a). Due to low sterol concentrations and co-elutions, particularly in the deeper mat layers, sterols were not quantified via peak integration in the total ion currents (TIC). Instead, the summed ion traces of $[m/z \; 129 + (M^+-90) + M^+]$ for the TMS-derivatives of $\Delta^5$- and $\Delta^{5,22}$-stenols, and $[m/z \; 215+ (M^+-90) + M^+]$ for the TMS-

derivatives of stanols were used. Appropriate correction factors were applied according to the response of these ions *vs*. concentration in the mass spectra of standard compounds. Average standard deviations of sterol concentrations were determined from repeated analyses of sample material. Hopanoids and fatty acids were identified based on the mass spectra and retention times.

**2.5 Pyrolysis-gas chromatography-mass spectrometry (Py-GC-MS)**

Aliquots of the decalcified extraction residues were pyrolysed on a fast-heating Pt-filament using a Pyrola 2000 pyrolysis device (Pyrolab SB) coupled to a Varian CP3800 GC and a Varian 1200L MS as described elsewhere (Shen et al., 2018a). An internal standard (*n*-eicosane D42, 120 ng) was routinely added to check the performance of the chromatographic system.

     Additionally, the Eocene Green River shale was used as a reference material (Eastern Utah, White River Mine, BLM Oil Shale Research, Development, and Demonstration Lease UTU-84087).

**2.6 Compound-specific stable carbon isotopes analysis**

Compound-specific stable carbon isotope ratios were measured for sterols and fatty acids in the freely extractable lipid fractions of the microbial mat. Analyses were conducted using a Thermo Scientific Trace gas chromatograph (GC) coupled to a Delta Plus isotope ratio mass spectrometer (IRMS). The conventional CuO/NiO/Pt reactor was used and combusted at 940°C. The GC-column used was an Agilent DB-5 coupled to an Agilent DB-1 (each 30 m length, 250 μm internal diameter,

and 0.25 μm film thickness). Lipid fractions were injected into a splitless injector and transferred to the GC column at

290°C. The carrier gas was helium at a flow rate of 1.2 ml/min. The temperature program for analyzing lipid fractions was ramped from 80°C, followed by heating to 325°C (at 5°C/min, held for 60 min). Analyses of laboratory standards were carried out to control the reproducibility of measuring conditions, and measurements were calibrated by using $CO_2$ gas of known isotopic composition.

5       *n*-Heneicosanoic acid and androstanol standards were analysed by GC-C-IRMS as non-derivatized lipids, and after derivatization as methyl esters (ME-) and trimethylsilyl (TMS-) ethers, respectively, to determine the carbon isotope values of the derivatizing groups. The $\delta^{13}$C-values of derivatized lipids were corrected for the additional carbon according to the equations provided in Goñi and Eglinton (1996).

## 3 Results

**3.1 General characterization of the microbial mat**

The microbial mat has a thickness of c.10 cm. Based on its macroscopic appearance, and on data from Blumenberg et al. (2015), the mat shows two major phases of development. The upper, younger growth phase is represented by layer 1 (photosynthetically active mat) and layer 2 (each c. 1 cm thick, Figs. 1, 2). These layers have a cohesive texture, sticking together when handled, due to abundant and relatively fresh organic material (i.e. EPS) of bright orange, green and brown

colours. Layer 1 includes small and scarce mineral precipitates, whereas layer 2 shows more abundant whitish minerals within its organic matrix (Fig. 2c). Layer 2 is underlain by a thin but distinctive, laterally continuous mineral crust (layer 3), which separates the two growth phases of the mat. Below the crust, the older growth phase is represented by layers 4, 5 and 6 (c. 7 cm thick in total, Figs. 1, 2). The lower layers are more friable than layers 1-2. They mainly show brown and beige colours and have a crumbly appearance, due to a higher abundance of mineral particles as compared to EPS (Fig. 2c). The

minerals observed within the mat layers are mainly aragonite ($CaCO_3$), with minor amounts of gypsum ($CaSO_4$) found only in the uppermost layer 1 (Shen et al., 2018b).

**3.2 Bulk geochemical data**

Bulk geochemical data for individual mat layers are shown in Table 1a. In the original non-decalcified mat, relatively high $C_{org}$ contents were observed in layers 1 and 2 (4.7 and 6.2 %, respectively; Table 1a and Fig. 3a), consistent with a more

fresh, cohesive appearance of the organic matrix in these layers. The earlier growth phase below showed constant $C_{org}$ contents < 2 %, with the lowest value found for layer 6 (1.2 %; see Table 1a). The $CaCO_3$ content of the mat increased significantly with depth (Fig. 3a; Table 1a). The lowest value was observed in the top layer 1 (27.1 %), a strong enrichment occurred in layer 2 (73.1 %), and constantly high contents (> 90 %) were found for all deeper mat layers. This is consistent with the observation of more abundant mineral precipitates downwards in the mat. The highest sulfur content was detected

for layer 1 (9.8 %), due to gypsum precipitates. Below, S decreased sharply (1.2 % in layer 2) and retained low values (<

%) in the earlier growth phase of the mat (~0.3-0.5 %). Nitrogen showed generally low contents (0.14-0.75 %) throughout the mat.

In the decalcified extraction residues (Table 1b), $C_{org}$ showed a broad range but increased significantly with depth, with the highest value observed in layer 6 (42.3 %; also see Fig. 3b). N was likewise enhanced in the deeper parts, with the highest amount found in layer 5 (6.7 %). By contrast, a decrease in S content was observed with depth, with highest values occurring in the topmost mat layer 1 (10.4 %).

### 3.3 Freely extractable lipids

#### 3.3.1 Freely extractable sterols

Various sterols were detected in the freely extractable lipid fractions, including saturated sterols (stanols; $C_{27}\Delta^0$, $C_{28}\Delta^0$, $C_{29}\Delta^0$) and unsaturated sterols (stenols; $C_{27}\Delta^5$, $C_{28}\Delta^{5,22}$, $C_{28}\Delta^5$, $C_{29}\Delta^{5,22}$, $C_{29}\Delta^5$; see Fig. 4; Table 2a). In addition, distinctive $C_{31}$ sterols were also detected and identified as 22,23-methylene-4α,23,24-trimethylcholest-5-en-3β-ol (4α-methylgorgosterol) and 22,23-methylene-4α,23,24-trimethylcholestan-3β-ol (4α-methylgorgostanol), respectively, based on the retention times and comparison with published mass spectra (Fig. S1; Atwood et al., 2014; Houle et al., 2019).

Figure 5a shows the variations of $C_{27}$-$C_{29}$ stenols *vs.* stanols in the freely extractable lipid fractions through the mat profile. The highest abundance of sterols occurred in the topmost layer 1 (26.05 µg/g dry mat, see Fig. 5a). Concentrations decreased drastically below, and remained low from layer 3 onwards. $C_{28}$ and $C_{29}$ sterols were the most dominant sterols in layer 1 while the $C_{31}$ sterols dominated in the deeper layers (Fig. S3). Unlike the other sterols, the $C_{31}$-sterols showed enhanced concentrations in layer 5, being three to ten times higher than in layers 3, 4 and 6 (Fig. 3c; Table 2a). In general, stenols were by about an order more abundant than stanols. Both groups showed highest concentrations in layer 1, and a major decrease within the mat (see Table 2a).

In the freely extractable lipids, the ratios of 5α-stanols to their corresponding $\Delta^5$-stenols (stanol/stenol ratios) showed no consistent trend within the profile (Fig. 6a; Table 3). The $C_{27}$-, $C_{28}$- and $C_{29}$- stanol/stenol ratios increased in the upper, younger growth phase of the mat, with the highest value observed for layer 3, but decreased again in the deeper, older growth phase (Fig. 6; Table 3). In contrast, stanol/stenol ratios for the $C_{31}$ sterols declined from layer 1 to layer 5, and showed a remarkable increase in layer 6.

A reliable compound-specific $\delta^{13}C$ value could be obtained for the coeluting $C_{31}$-sterols from the freely extractable lipids in layer 1. These compounds showed strong enrichments in $^{13}C$ ($\delta^{13}C$ = -7.2 ‰). Fatty acids (including $C_{14}$-$C_{19}$ homologues) showed similarly high $\delta^{13}C$ values ranging from -4.4 to -11.7 ‰.

#### 3.3.2 Freely extractable hopanoids

Several GC-amenable hopanoids were detected in the freely extractable lipids, with major compounds being hop-22(29)-ene (diploptene), ββ-bishomohopanoic acid and ββ-bishomohopanol. The summed major hopanoids showed highest abundances

in layer 1 and 2 (about 15 µg/g dry mat). Below, hopanoid concentrations sharply decreased to < 4 µg/g dry mat in layers 3 and 4, but returned to moderate values in the deeper layers 5 and 6 (see Table S3, Fig. 7a).

### 3.3.3 Freely extractable fatty acids (FAs)

FAs in the freely extractable lipid pool show carbon numbers ranging from 14 to 30 (Table S1). Short-chain FAs ($C_{14}$-$C_{19}$) are predominant, making up > 90 % of the total in the upper layers and 50-60 % in the deeper layers. Further, medium-chain FAs ($C_{20}$-$C_{23}$) occur in low abundances (< 10 %) throughout the mat profile. Long chain FAs ($C_{24}$-$C_{30}$) made up only a few % of the freely extractable lipids in the upper three layers. However, the relative abundance of these long-chain FAs significantly increased in the deeper part of the mat, with c. 40 % observed for layer 5. Saturated and mono-unsaturated $C_{16}$ and $C_{18}$, and a cyclopropyl-FA (tentatively identified as *cis*-9,10-methyleneoctadecanoic acid) were the dominant short chain FAs. In addition, terminally branched *iso-/anteiso-* $C_{15}$- $C_{17}$ FAs were detected, showing a decreasing trend with depth (Table S1). With respect to the total FAs, highest concentrations were found in layer 1 (212.72 µg/g dry mat), and lowest in layer 3 (3.34 µg/g dry mat). Below layer 3, FA concentrations slightly increased again with depth (see Table S3).

### 3.3.4 Depth distributions of freely extractable compounds

Depth distributions of summed GC-amenable hopanoids, FAs, and steroids in the free lipids are presented in Fig. 7a. In all layers, FAs were by an order more abundant as hopanoids and steroids, but the relative distributions of the three compound classes resembled each other (high amounts in layer 1, low amounts in layers 3 and 4, Table S3). Quite similar distributions were observed when the compound concentrations were plotted against $C_{org}$, due to relatively low organic carbon throughout the mat (1.20-6.23%, Table 1a, Fig.7a).

## 3.4 Carbonate-bound lipids

### 3.4.1 Carbonate-bound sterols

Carbonate-bound sterols include $C_{27}$-$C_{29}$ conventional sterols as well as $C_{31}$ sterols. When compared to the freely extractable lipids, their concentrations are low, particularly in the surface layers (Fig. 5b; Table 2b). In the topmost layer 1, carbonate-bound sterols were virtually absent. In layer 2, carbonate-bound sterols occurred, but were still much less abundant as freely extractable sterols. In the deeper mat layers (3-6), however, the carbonate-bound and freely extractable sterols were in the same order (~$10^{-2}$ µg/g dry mat range). $C_{27}$-$C_{29}$ sterenes were also detected in both lipid pools, but only at trace abundances.

$C_{31}$-sterols were the most abundant sterols in both fractions (ranging up to ca. 85 % in the deeper part of the mat), followed by $C_{29}$-sterols (Fig. S3). The concentration of carbonate-bound $C_{31}$-sterols increased in the bottom layer 6, which is distinguished from the other sterols (Fig. 3d). The stanol/stenol ratios in the carbonate-bound lipids increased for the $C_{27}$-$C_{29}$ pairs between layers 1 and 3, and again decreased further downwards, thus being similar to the freely extractable lipids (see

Fig. 6b; Table 3). No stanol/stenol ratios could be obtained for the carbonate-bound $C_{31}$ sterols, as carbonate-bound $C_{31}$ stenols were virtually absent throughout the mat.

### 3.4.2 Carbonate-bound hopanoids

Small amounts of carbonate-bound hopanoids (mainly hop-17(21)-ene and ββ-bishomohopanoic acid) were observed in all mat layers except layer 1 (Table S2). The concentrations showed no consistent trend with mat depth. In layers 2, 5, and 6, the summed hopanoids were by an order lower than in the free lipids (<1 µg/g dry mat; see Table S4). In contrast, the amount of carbonate-bound hopanoids was markedly enhanced in layer 3 (3.22 µg/g dry mat), thus being similar to the freely extractable lipids (Table S3).

### 3.4.3 Carbonate-bound fatty acids (FAs)

Carbonate-bound FAs range in chain length from $C_{14}$-$C_{28}$, with short-chain saturated and mono-unsaturated homologues predominating ($C_{16}$ - $C_{19}$). Apart from the straight chain (*n*-) FA, *iso-/anteiso*-branched $C_{15}$ - $C_{17}$ FA and a cyclopropyl-FA (tentatively identified as *cis*-9,10-methyleneoctadecanoic acid) were observed (Table S2). The summed concentrations of carbonate-bound FAs were in the same order as in the freely extractable lipid fraction ($10^0$~$10^1$ µg/g dry mat range). An exception was found for layer 1, where carbonate-bound FAs were an order less abundant ($10^2$ µg/g dry mat range; Table S3 and S4). The depth distribution of the summed carbonate-bound FAs showed an increase until layer 3 (highest concentration: 80.45 µg/g dry mat), followed by a sharp decrease in the deeper parts, with the lowest values observed for layer 5 (6.87 µg/g dry mat; Table S4).

A unique feature of the carbonate-bound FA fractions is the occurrence of saturated α,ω-dicarboxylic acids (α,ω-diacids) ranging in carbon numbers from 21 to 28. These diacids were detected in low concentrations (< 1 µg/g dry mat) only in layers 3 and 4, where they make up < 10% of the carbonate-bound FAs.

### 3.4.4 Depth distributions of carbonate-bound compounds

The depth distributions of the summed carbonate-bound compounds (Fig. 7b) revealed much (by an order) higher concentrations of FAs as compared to steroids and hopanoids. Whereas carbonate-bound steroids displayed no significant changes throughout the profile, summed carbonate-bound FAs and hopanoids showed a decreasing trend with depth, interrupted by a remarkable enrichment in layer 3 (Table S4). Largely identical distributions were observed when these compound classes were plotted against $C_{org}$ (Fig.7b).

## 3.5 Decalcified extraction residues

Ion chromatograms representing steroids and hopanoids released by pyrolysis of the decalcified extraction residues are shown in Fig. S2. Throughout the mat, steroids were not observed in the pyrolysates, indicating that kerogen-bound steroids

released from the macromolecular fraction were below our Py-GC-MS detection limit (~1 ng per analyte, see Fig. S4). On the other hand, hopanoid moieties were pyrolysed from the insoluble matter of each mat layer. Whereas only traces were detected in the pyrolysates of layer 1, their abundance showed a clearly increasing trend with mat depth (Fig. S2).

## 4 Discussion

### 4.1 Origin of sterols

The studied mat contained a broad variety of $C_{27}$-$C_{29}$ sterols as well as two $C_{31}$ sterols, indicating potential sources like animals, fungi, algae and terrestrial plants (Atwood et al., 2014; Houle et al., 2019; Volkman, 1986; Volkman, 2003). The concentrations of freely extractable sterols in the topmost layer 1 in the studied Lake 2 mat are similar to Lake 2A and Lake 22 (~$10^2$-$10^3$ µg/g $C_{org}$; Bühring et al., 2009; Shen et al., 2018a; see Fig. 5a). However, sterols in the deeper layers are much less abundant in the Lake 2 mat as compared to other mats in Kiritimati lakes.

Figure S3 shows the relative distribution of summed $C_{27}$- vs. $C_{28}$- vs. $C_{29}$- vs. $C_{31}$- sterols in the microbial mat layers. In both lipid fractions, the $C_{31}$-sterols are predominant, suggesting inputs from dinoflagellates (Atwood et al., 2014; Houle et al., 2019). $C_{29}$-sterols make up the next most abundant group of sterols, potentially indicating contributions from either algae or terrestrial plants (Volkman, 1986); moreover, these compounds are known to be produced by some algae, including diatoms (Rampen et al., 2010; Volkman, 2003).

The high $\delta^{13}C$ value of -7.2 ‰ for the $C_{31}$-sterols, as well as similarly high values measured for fatty acids from layer 1, imply that the carbon source of these compounds was autochthonous and derived from the hypersaline, $CO_2$-limited ecosystem of Lake 2 (cf. Schouten et al., 2001). Previous work on carbon isotope compositions of sterols in a mat from the adjacent Kiritimati Lake 2A showed $\delta^{13}C$ values from -19 to -23 ‰ (Bühring et al., 2009). In addition, Trichet et al. (2001) reported $\delta^{13}C$ values for sedimentary bulk OM from -14 to -17 ‰ in Kiritimati Lake 30. Thus, both studies showed more depleted values than those observed for Lake 2. An explanation could be a better $CO_2$ exchange in those lakes, due to their shallow water layer (a maximum depth of 0.2 m in Lake 2A, Bühring et al., 2009; depth of 0.9 m in Lake 30, Trichet et al., 2001), leading to the relatively light $\delta^{13}C$ signatures. Another explanation could be that shrinking lake water bodies caused by La Niña dry events are often associated with massive increases in lake salinities (Trichet et al., 2001). For instance, Lake 2A (Bühring et al., 2009) was observed to be nearly dried out during our sampling campaign in 2011. The increasing salinities may result in a $CO_2$-limited ecosystem, leading to enrichment in $^{13}C$. For Lake 2, such reinforced $CO_2$-limitation is not only supported by the high $\delta^{13}C$ values of individual biomarkers, but also by $\delta^{13}C$ values of carbonates that were reported to be as high as +6 ‰ (Arp et al., 2012).

## 4.2 Taphonomy of lipids

### 4.2.1 Freely extractable lipids

The sterols in the studied mat are probably sourced from plankton or organisms thriving at the mat surface, because eukaryotes are generally depending on an oxygenated environment and would not thrive in anoxic, deeper parts of the mat. The abundance of total extractable sterols was high in the top mat but significantly decreased (up to > 90%) immediately below the topmost layer 1, and kept at trace amounts ($10^{-1}$ μg/g dry mat, Fig. 5a) in the deeper part of the mat. Whereas the possibility of a change in microbial vs eukaryotic input over the time of mat deposition has to be considered, there are no major changes in the texture of the carbonate phases of the mat (except the thin mineral crust representing layer 3), which would suggest major environmental changes leading to an exclusion of eukaryotes. We therefore interpret this substantial decrease to result from major sterol degradation caused by heterotrophic microorganisms, thus suggesting the existence of a major "mat-seal effect" in the mat studied.

It can furthermore be expected that most sterols were initially introduced as stenols. Subsequent alteration by early diagenetic processes within the mat would have resulted in a variety of sterol transformation products. Reduction of $\Delta^5$-stenols to 5α-stanols (hydrogenation) is a known result of anaerobic microbial degradation (Rosenfeld and Hellman, 1971; Wakeham, 1989). Consequently, stanol/stenol ratios may reflect the extent of microbial alteration under anoxic conditions (i.e. under low redox potential; Gaskell and Eglinton, 1975; Nishimura, 1977; Wakeham, 1989). Several investigations have reported such conversion in microbial mats (Grimalt et al., 1992; Scherf and Rullkötter, 2009; Słowakiewicz et al., 2016), including some mats from other lakes on Kiritimati (Bühring et al., 2009; Shen et al., 2018a).

Stanol/stenol ratios for $C_{27}$-$C_{29}$ pairs in the free lipids initially increased with depth as expected, and showed highest values in layer 3, indicating low redox potentials and a pronounced anaerobic microbial transformation of stenols therein (Fig. 6a). In the deeper layers (4-6), however, ratios decreased again. We interpret this to result from a more efficient microbial OM degradation under higher redox potentials prevailing during the more rapid accretion of the earlier growth phase of the mat (Blumenberg et al., 2015). This idea is supported by constantly lower $C_{org}$ contents in the earlier growth phase (Table 1a). As an exception, the $C_{31}$-stanol/stenol ratios showed an outstanding behaviour as they steadily decreased with depth, but sharply increased again in the bottom layer 6. Possibly, primary input variations played a role for the distributions of these compounds. 4α-methylgorgostanol has been reported in a few dinoflagellate species belonging to the genera *Peridinium*, *Alexandrium* and *Pyrodinium*, (Atwood et al., 2014; Houle et al., 2019, and refs therein). The partly co-eluting stenol has been tentatively identified as 4α-methylgorgosterol which has been reported in resting cysts but not in the motile cells of the dinoflagellate *Peridinium umbonatum* var. *inaequale* (Amo et al., 2010). Consequently, the $C_{31}$ sterols observed may partly derive from sedimentary resting cysts and may have been less affected by microbial recycling than conventional $C_{27}$-$C_{29}$ sterols. It may also be speculated that the unusual side-chain structure and methylation pattern of 4α-methylgorgosterols hamper enzymatic microbial degradation (e.g. Giner et al., 2003, and refs therein). The steadily

increasing relative abundances of $C_{31}$- *vs.* $C_{27}$-$C_{29}$ sterols in the mat profile (Fig. S3) suggest that $C_{31}$ sterols experienced different degradation patterns as compared to conventional sterols.

The GC-amenable hopanoids observed may have largely formed as earliest (eogenetic) products of bacteriohopanepolyols (BHPs; Rohmer, Bouvier-Nave and Ourisson, 1984) via progressive, microbially driven defunctionalisation. In the top layers 1 and 2, these hopanoids showed the highest concentrations within the mat (Fig. 7a). Below, they significantly decreased, suggesting major anaerobic degradation or, alternatively, binding to macromolecules. In the deeper layers 5 and 6, however, concentrations of hopanoids increased again to moderate values. While it can be assumed that a part of the initially produced BHPs has been transformed into the GC-amenable hopanoids observed. Another part has evidently been incorporated into macromolecular organic matter, as revealed by the release of hopanoids from the decalcified extraction residues by Py-GC-MS (Fig. S2). Steroid/hopanoid ratios show a significant drop in the upper two layers (possibly due to a major biodegradation of steroids therein), whereas the ratios keep fairly constant in the deeper mat layers, suggesting a similar degradation of both steroids and hopanoids at depth. However, it should be considered that different input/degradation patterns of $C_{31}$ sterols (see above) may have influenced the steroid/hopanoid ratios observed.

To further check for the sources and additional input of lipids, we observed the FA distributions in the mat profile (Table S1 and S3). The predominance of saturated and mono-unsaturated $C_{16}$ and $C_{18}$ FAs, along with a cyclopropyl-$C_{19}$ FA reveal major contributions from bacteria (Kates, 1964; Kaneda, 1991; Taylor and Parkes, 1985), whereas low amounts of homologues $> C_{20}$ indicate only minor allochthonous inputs derived from higher plant lipids (Brassell et al., 1980; Cranwell, 1982). Fairly constant steroid/FA ratios in layers 1 to 4 (Fig. 7a and Table S3) suggest that both compound classes experienced similar preservation/degradation pathways. Again, there is little evidence for additional production of microbial lipids in deeper parts of the mat. Rather, and unexpectedly, steroid/FA ratios even increased in the deepest part of the mat (see Table S3), which is likely due to additional inputs of resistant $C_{31}$-sterols derived from dinoflagellate lipids, as discussed above.

*4.2.2 Carbonate-bound lipids*

Unlike freely extractable sterols, carbonate-bound sterols were virtually absent at the top of the mat, and likewise showed constantly low abundances below (Fig. 5b). These observations suggest that the carbonate matrix played no important role in encasing (i.e., preserving) sterols in this mat and may be taken as indicative of a minor role of eukaryotic organisms, and their OM, in carbonate formation. In contrast, studies revealed that distinctive microbial lipids preserved in the carbonate matrix reflect a constructive role of their source organisms in carbonate formation and/or their continuous incorporation during the precipitation processes (Peckmann and Thiel, 2004; Summons et al., 2013; O'Reilly et al., 2017).

Stanol/stenol ratios for the carbonate-bound $C_{27}$-$C_{29}$ sterols were similar as in the free lipids, with the highest value observed in layer 3, which is comprised of a dense mineral crust (Fig. 6b; see section 4.2.1). At the same time, both

carbonate-bound hopanoids and microbial FAs were remarkably enriched in layer 3, indicating intensive microbial activity and, seemingly, an enhanced preservation of prokaryotic lipids in the carbonate matrix of this layer.

Steroid/hopanoid ratios in the carbonate-bound fraction showed no consistent trend through the depth profile. After a major drop in layer 3, due to the above mentioned increase in hopanoids, steroid/hopanoid ratios increased again in the lower

layers 5 and 6. This may result from a higher abundance of carbonate-bound $C_{31}$-sterols in the bottom layers (as it was also observed in the freely extractable sterol fraction, see above). Steroid/FA ratios in the carbonate-bound lipids showed very low values through the mat profile. Again, the lowest values were observed for layer 3, due to the highest concentration of FAs observed therein.

A unique feature in the carbonate-bound lipids is the occurrence of α,ω-diacids. Previous work showed that these lipids

may have multiple biological sources, e.g., higher plants (Kolattukudy, 1980) and sea-grass (Volkman et al., 1980). Given the presence of our findings of terrestrial biomarkers (albeit in low abundance) in the studied mat, these α,ω-diacids could be sourced from higher plant waxes. On the other hand, α,ω-diacids may also be foming *in situ*, for instance via terminal oxidation of monoacids or other aliphatic moieties such as *n*-alkanes (Ishiwatari and Hanya, 1975; Johns and Onder, 1975). Interestingly, α,ω-diacids were also reported in a recent study on Cretaceous hydrocarbon seep limestones, where they were

only detected after the dissolution of the authigenic carbonate minerals (Smrzka et al., 2017). Likewise, α,ω-diacids were reported to be remarkably more abundant in carbonate concretions than in their clastic host rocks (Thiel and Hoppert, 2018). Taken together, these results might indicate that the formation of α,ω-diacids is directly associated with carbonate precipitation, or that these compounds are better preserved in carbonate matrices.

*4.2.3 Decalcified extraction residues*

Steranes have previously been detected in the insoluble macromolecular OM of benthic mats using catalytic hydropyrolysis (HyPy) (Blumenberg et al., 2015; Lee et al., 2019). In hydropyrolysates of microbial mats from Guerrero Negro in Mexico, concentrations of steranes and hopanes were in the same range (Lee et al., 2019). Likewise, previously reported HyPy data for our mat from Lake 2 (Blumenberg et al., 2015) also showed the presence of both, hopanoids and steroids, but the latter were more than 20 times less abundant. This finding of predominant hopanoids in the Lake 2 mat is concordant with our Py-

GC-MS data, however, steroid moieties were even below detection limit in our analyses (Fig. 2S). The non-detection of steroids here could be due to a lower detection limit in our Py-GC-MS setup, where an absolute amount of >1 ng of the target analyte is required to obtain an interpretable mass spectrum (according to analyses of reference compounds). Further, unlike in HyPy, there is no possibility in Py-GC-MS for a downstream chromatographic separation and concentration of the analytes.

Taken together, the strong decline in freely extractable sterols below the uppermost mat layer, along with only minor incorporation into carbonate and macromolecular OM, suggest that major degradation of steroids occurred during eogenesis and earliest diagenesis in the Lake 2 mat studied.

## 4.3 Comparison with sterol taphonomy in other microbial mats

Major differences are evident between the depth distributions of steroids in the Lake 2 mat studied here and those reported previously from a Lake 22 mat (Shen et al., 2018a). Whereas the sterol concentrations in the topmost layers are similar in both mats ($10^2$ µg/g $C_{org}$ range), the Lake 22 mat showed no systematic decrease in sterols with depth. Such entirely different
behaviour of sterols in the mats from the two adjacent lakes raises questions about potential mechanisms causing the observed variation.

One explanation for the differences observed could be differences in salinity. In 2011, the salinity of Lake 22 (Shen et al., 2018a) was 250 ‰, whereas Lake 2 showed only 125 ‰. High salinity may reduce microbial cell growth and reproduction, and limit the metabolism of microorganisms. The resulting decrease in bacterial activity would affect the
biodegradation rates of organic compounds (Abed et al., 2006). Several studies reported that the degradation rates of hydrocarbons significantly decrease as salinity increases (Abed et al., 2006; Ward and Brock, 1978). In turn, lower salinity supports the proliferation of a more diverse microbial community (Bolhuis et al., 2014), thus possibly enhancing OM biodegradation. As a consequence, conditions for heterotrophic microorganisms could be more favourable in Lake 2 as compared to the extremely hypersaline Lake 22, thus accelerating the biodegradation rates of organic molecules, including
sterols.

A second plausible explanation for the differences between the Lake 2 and 22 mats might be associated with the environmental properties of both lakes, particularly water depth. A major drought period prevailed in Kiritimati from 2002 to 2011, as a result of a very strong La Niña dry event. Due to reduced rainfall, the water level of the lakes in Kiritimati generally dropped, so that, in some areas, parts of the lake bottoms became subaerially exposed (this has been observed for
Lake 2A; see section 4.1). The Lake 22 microbial mat was collected at the margin of the lake (Fig. 2f; water depth c. 0.2 m; Shen et al., 2018a). Therefore, mats from this shallow sampling site may have suffered from heavy evaporation due to such major drought events. Indeed, the Lake 22 mat studied by Shen et al. (2018a) showed an irregular top layer of V-shaped fractures, which are characteristic features of subaerial exposure in evaporitic settings. On the other hand, the Lake 2 mat studied here was collected in the lake centre at the water depth of 4 m and would have been clearly less prone to subaerial
exposure. These interpretations are further supported by other studies highlighting the influence of water depth and salinity on the microbial and biomarker composition of microbial mats (Pagès et al., 2014).

Likewise, much higher stanol/stenol ratios in the Lake 22 mat (Shen et al., 2018a) indicate a more intense anaerobic microbial transformation (yet no degradation) as compared to the Lake 2 mat studied here (Fig. 6). Whereas sterols in Lake 22 mat experienced major microbial *transformation* (stenols => stanols => sterenes), sterols in Lake 2 seemingly suffered
from major *degradation* that largely eliminated the primary eukaryotic signal. The contrasting distributions observed suggest that sterols have a higher preservation potential in microbial mats under stronger salinities and/or more desiccated conditions, such as those of Lake 22. Our finding of such significant differences in two adjacent mat settings on the same island should be considered when making generalizations for the fossil record from studies of sterols in modern microbialites

or microbial mats. Sterol preservation within microbial mats appears to be a complex process that may be strongly influenced by environmental parameters. Therefore, palaeoenvironments must be thoroughly constrained if the presence, or absence, of these compounds is interpreted in the study of ancient deposits.

## 5 Conclusion

The preservation of primary eukaryotic sterols and their progressive alteration was studied in a c. 1500 years old microbial mat from the hypersaline Lake 2 on Kiritimati. Conventional $C_{27}$-$C_{29}$ sterols decreased severely with depth, suggesting a progressive biodegradation of these compounds within the mat. A different pattern was observed for unusual, isotopically heavy $C_{31}$-sterols (4α-methylgorgosterol and 4α-methylgorgostanol; $\delta^{13}C$ = -7.2 ‰), which showed increasing abundances in the deeper mat layers. This may be explained by an enhanced resistance of these sterols against degradation, possibly due to

their unusual side-chain and/or an origin from highly resistant dinoflagellate resting cysts. Separate analysis of decalcified samples revealed that no significant 'trapping' of sterols into the mineral matrix occurred in this mat. Further, Py-GC-MS of decalcified extraction residues showed steroids to be below detection limit, in contrast to hopanoids which occurred abundantly in the pyrolysates throughout the mat profile. Our combined data suggest that the studied mat might have formed an effective filter against the preservation of sterols in the sedimentary record. For the studied mat, the results thus support

the hypothesis of a 'mat-seal effect' describing the degradation of eukaryote-derived lipids in benthic microbial mats. Our results are markedly different from those recently reported from another microbial mat from close-by Lake 22 where sterols showed no systematic decrease with depth. In that mat, higher salinity or temporal subaerial exposure have probably hampered microbial metabolism and promoted the preservation of steroids over degradation. The data combined in this study show that sterol taphonomy may strongly vary between different mat systems, and even contrasting sterol degradation

patterns may be expected in response to environmental conditions.

*Code and data availability.* Data can be found in the Supplement or can be requested from Yan Shen (yshen@gwdg.de).

*Author contributions.* YS had the main responsibility for analysing the data and writing the manuscript. JR and VT designed the project. JR conducted the field works, collected all the sample materials and participated in the writing process. VT contributed the conceptualisation of this work, participated in constructing the measurement setup and the writing process.

5    PSG had the main responsibility for the bulk data measurements, participated in the writing process. SWP participated in the interpretation of the data and the writing process. All the co-authors contributed to this work.

*Competing interests.* The authors declare that they have no conflict of interest.

*Acknowledgements.* We thank Prof. Dr. Gernot Arp and Dr. Martin Blumenberg for helpful information and constructive comments. Dr. Andreas Reimer is acknowledged for hydrochemical measurements. We also thank Dr. Jens Dyckmans for

10   performing the compound-specific carbon isotope measurements. Wolfgang Dröse, Birgit Röring and Dorothea Hause-Reitner are kindly acknowledged for laboratory assistance. Dr. Pablo Suarez-Gonzalez acknowledges funding by a postdoctoral research fellowship of the Alexander von Humboldt Foundation. This research has received funding from the German Research Foundation (DFG, Project Re 665/18-2 and Research Unit 571 "Geobiology of Organo- and Biofilms").

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

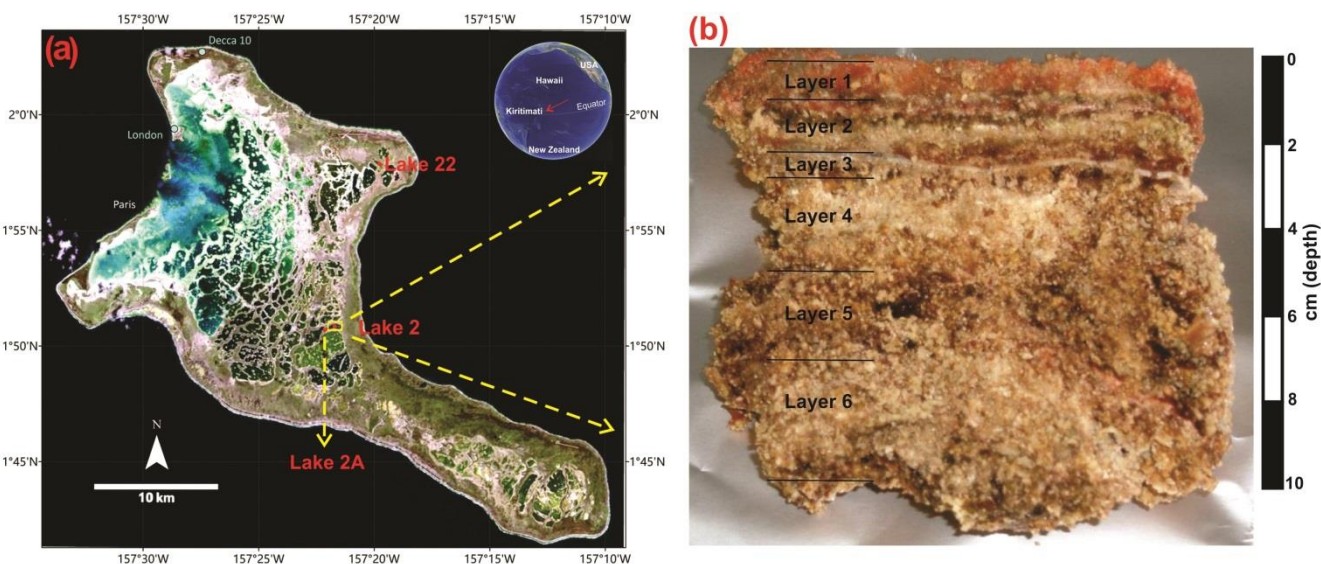

**Figure 1.** (a) Location of Kiritimati atoll in the Pacific Ocean and satellite view (Landsat 7 image, 1999) showing reticulate distribution pattern of the lakes (red dots: Lake 2 studied in this work; Lake 2A and 22 previously studied by Bühring et al., 2009 and Shen et al., 2018a); (b) the microbial mat sample from Lake 2 studied in this work.

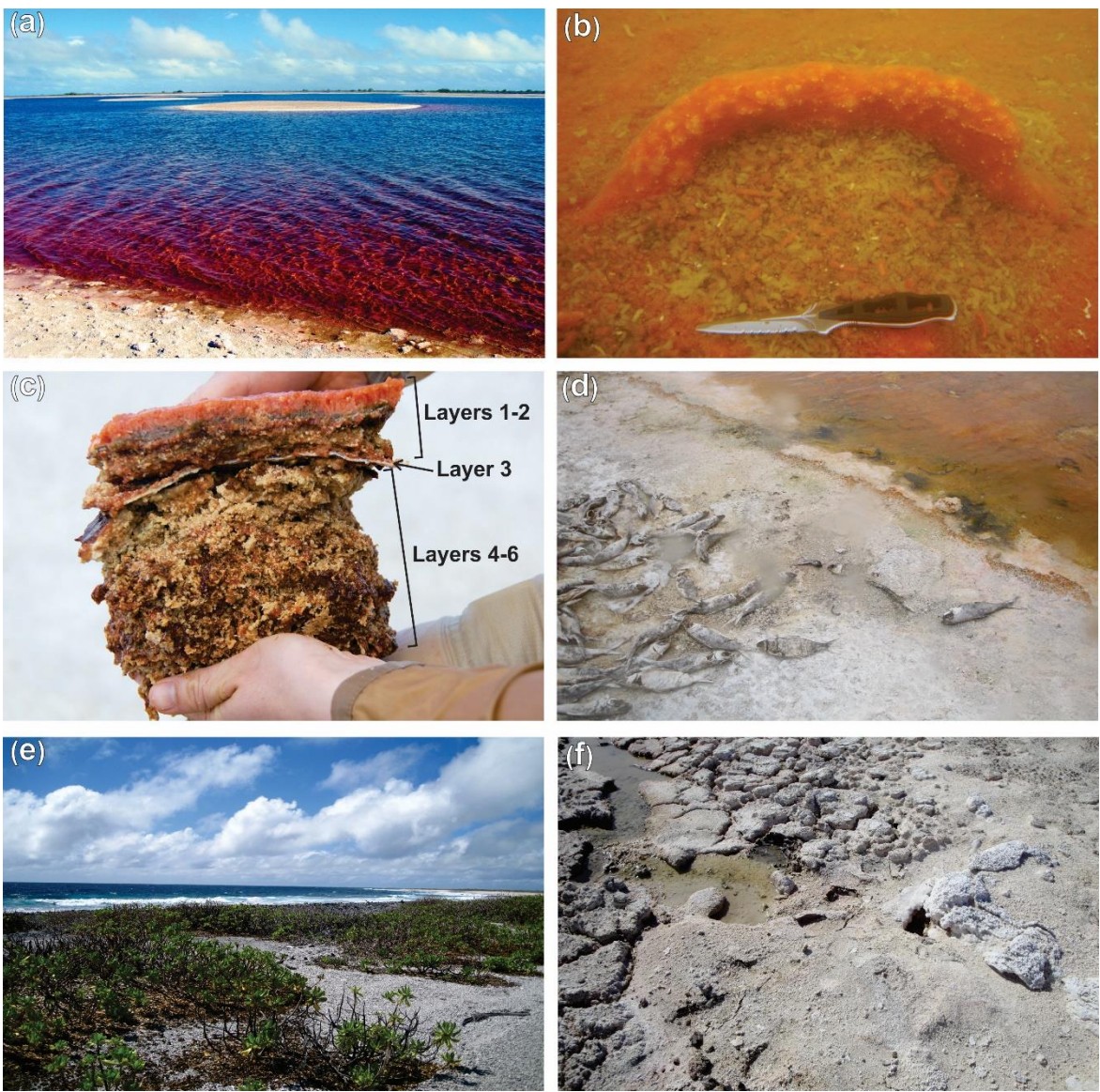

**Figure 2.** Field images: (a) general view of hypersaline Lake 2 in Kiritimati; (b) underwater photograph showing an example of a currently-active, orange-coloured microbial mat at the bottom of the lake; (c) the microbial mat sampled for this study, with clear colour-zonation; note the whitish mineral crust (Layer 3) separating the upper younger growth phase from the older, more mineralized layers; (d) lake shore showing dead fish; (e) vegetation around the lake area; (f) sampling site for hypersaline Lake 22 mat (Shen et al., 2018a).

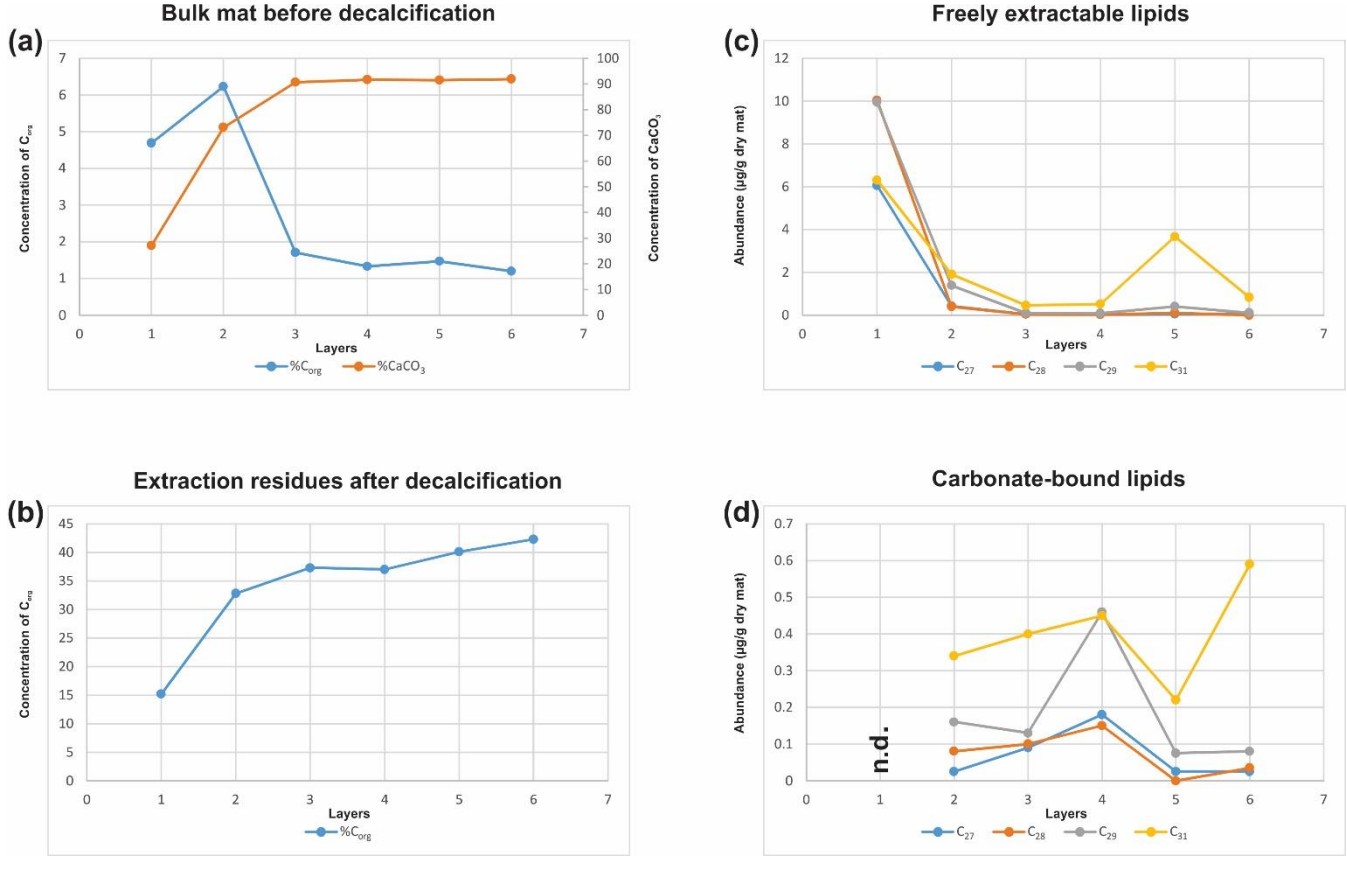

**Figure 3.** (a) $C_{org}$ and $CaCO_3$ contents of the bulk mat ($\%_{wt}$); (b) $C_{org}$ content of the extraction residues after decalcification ($\%_{wt}$); (c) Distribution of $C_{27}$- vs. $C_{28}$- vs. $C_{29}$- vs. $C_{31}$- sterols in the freely extractable lipids ($\mu g/g$ dry mat); (d) Distribution of $C_{27}$- vs. $C_{28}$- vs. $C_{29}$- vs. $C_{31}$- sterols in the carbonate-bound lipids of the microbial mat ($\mu g/g$ dry mat).

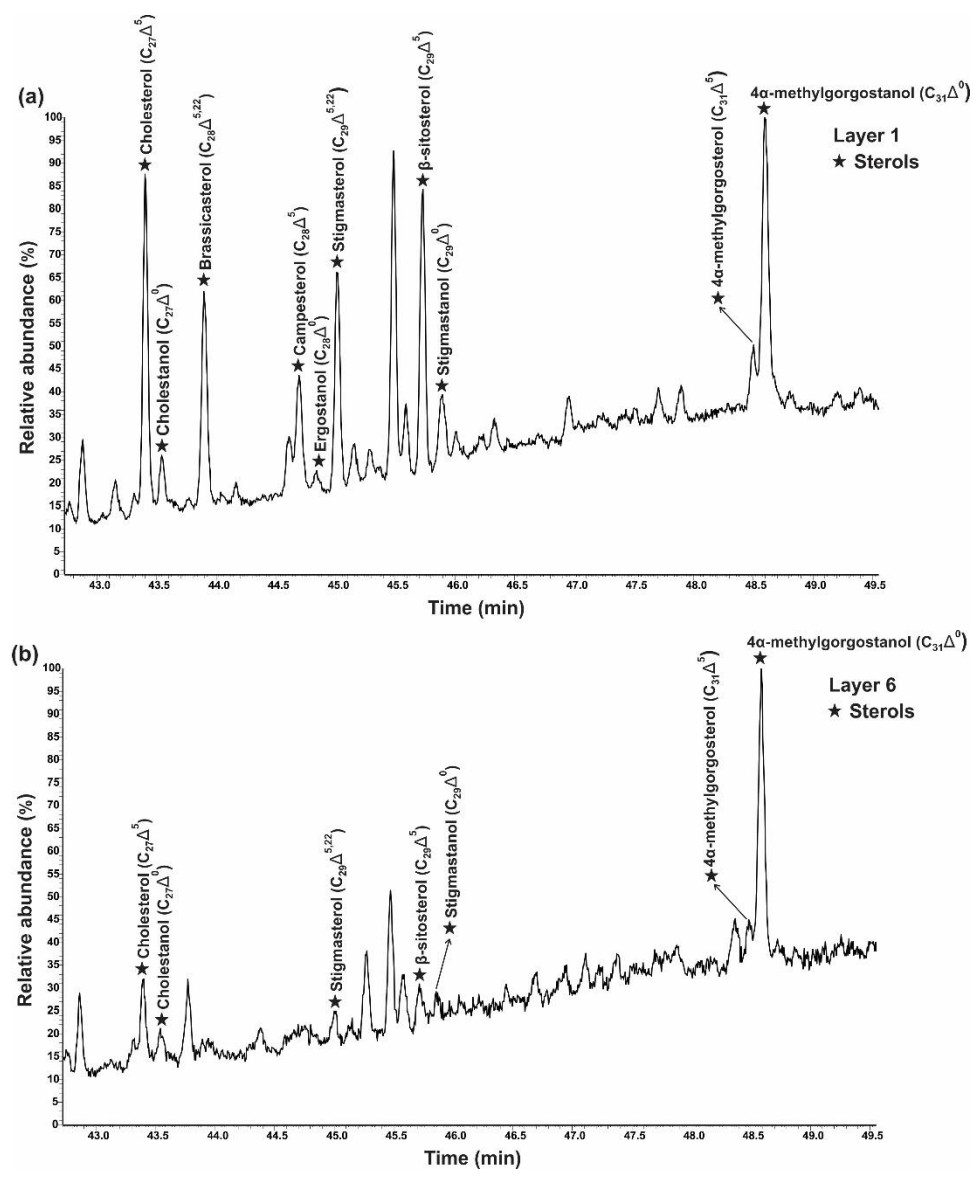

**Figure 4.** Partial GC-MS chromatograms (total ion current) show the distributions of freely extractable sterols (TMS-derivatives) in (a) layer 1, and (b) layer 6 of the microbial mat.

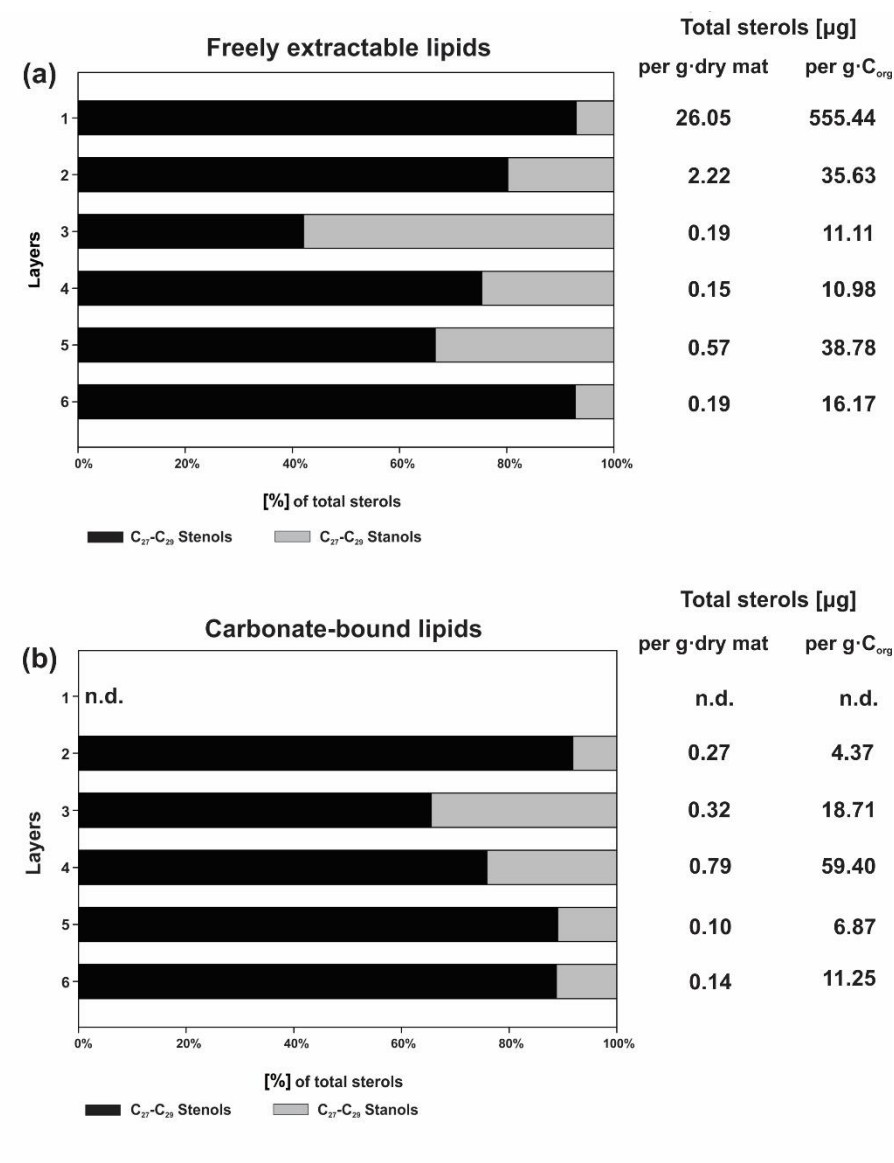

**Figure 5.** Distributions and concentrations of $C_{27}$-$C_{29}$ sterols in the microbial mat layers, (a) freely extractable lipids, and (b) carbonate-bound lipids.

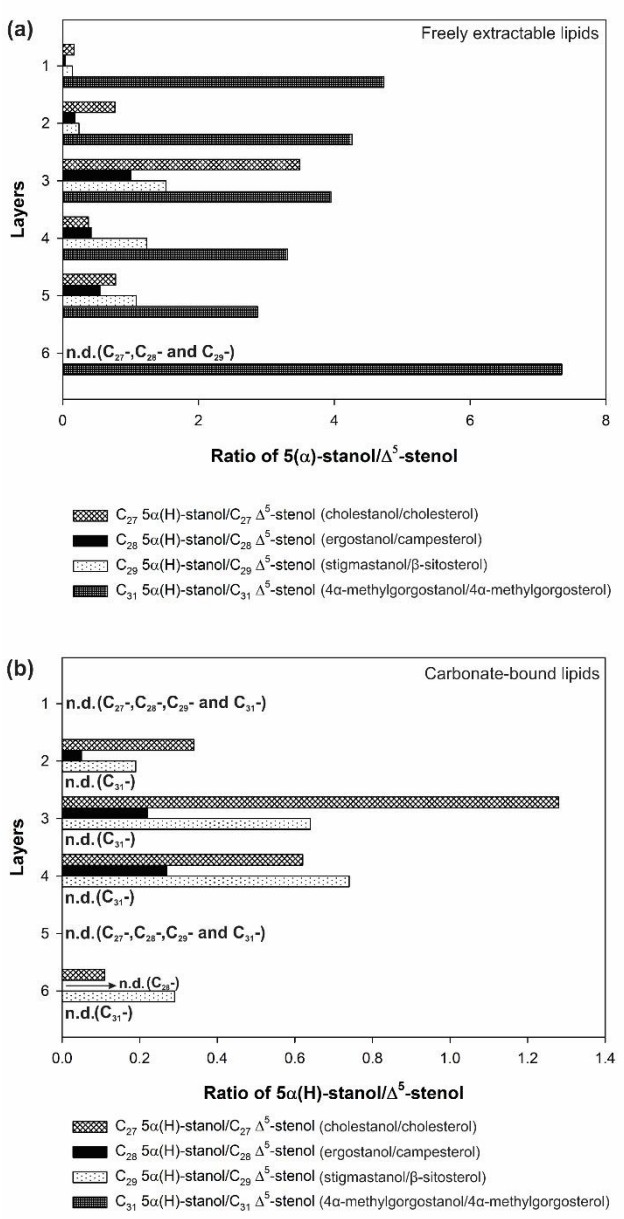

**Figure 6.** Stanol/stenol ratios for the microbial mat layers, (a) freely extractable lipids, and (b) carbonate-bound lipids.

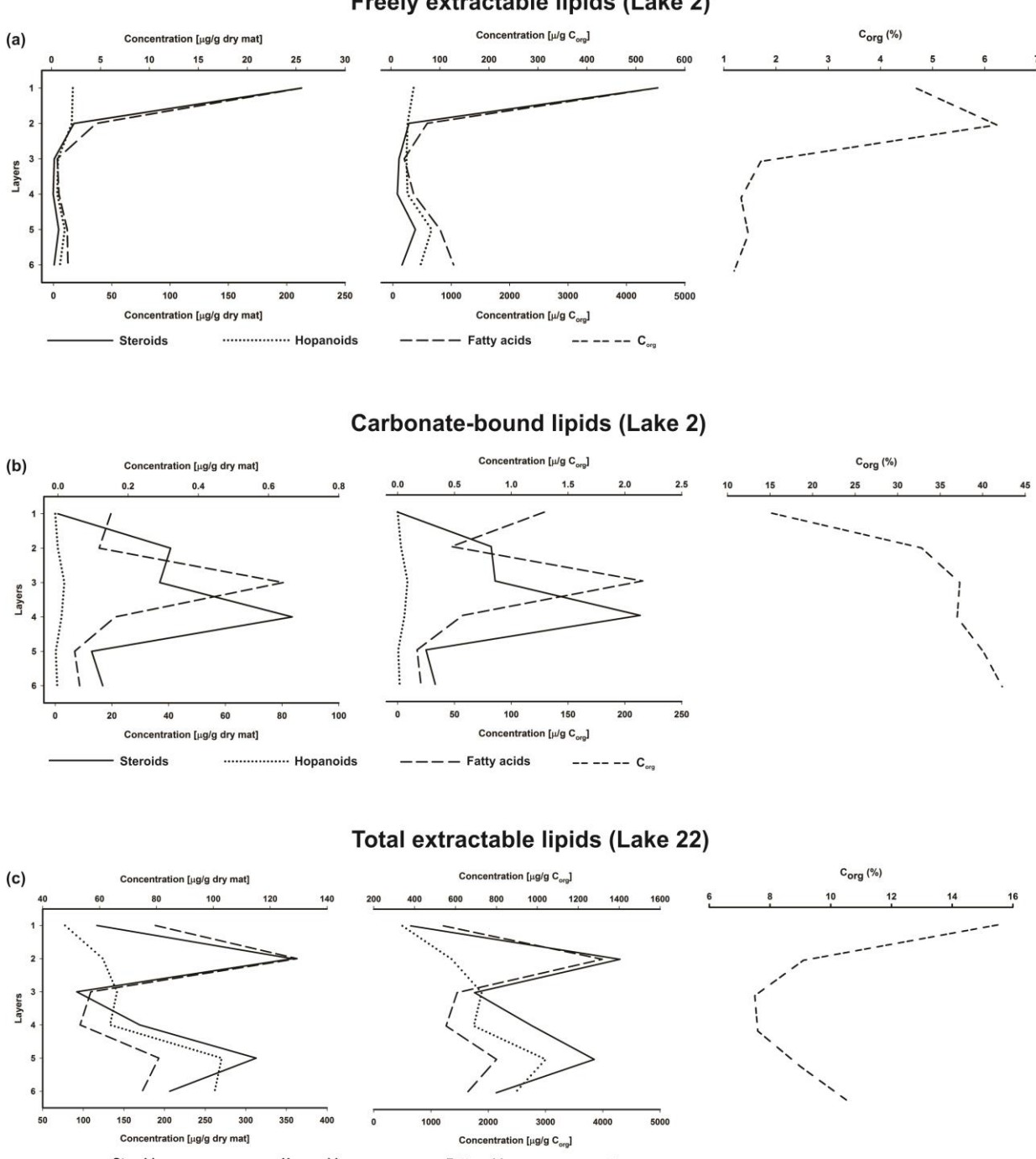

**Figure 7.** Depth distributions of steroids, hopanoids and fatty acids (µg/g dry mat; µg/g $C_{org}$), and $C_{org}$ (wt. %) in the microbial mat layers, (a) freely extractable lipids of Lake 2 mat, (b) carbonate-bound lipids of Lake 2 mat, and (c) total extractable lipids of Lake 22 mat (Shen et al., 2018a) (upper X axis of (a), (b) and (c) is applied to steroid concentrations, while lower X axis is applied to hopanoids and fatty acids).

**Table 1a**. Bulk geochemical data for the microbial mat (original mat layers before decalcification)

| Layers | $C_{tot}$ (%) | $C_{org}$(%) | $C_{carb}$ (%) | $CaCO_3$ (%) | $N_{tot}$ (%) | $S_{tot}$ (%) | $C_{org}/N$ | $C_{org}/S$ |
|--------|---------------|--------------|----------------|--------------|---------------|---------------|-------------|-------------|
| 1 | 7.94 | 4.69 | 3.25 | 27.10 | 0.41 | 9.78 | 11.40 | 0.50 |
| 2 | 15.00 | 6.23 | 8.77 | 73.10 | 0.74 | 1.21 | 8.40 | 5.20 |
| 3 | 12.59 | 1.71 | 10.88 | 90.70 | 0.16 | 0.33 | 10.50 | 5.10 |
| 4 | 12.33 | 1.33 | 11.00 | 91.70 | 0.19 | 0.49 | 7.00 | 2.70 |
| 5 | 12.45 | 1.47 | 10.98 | 91.50 | 0.20 | 0.52 | 7.30 | 2.80 |
| 6 | 12.23 | 1.20 | 11.03 | 91.90 | 0.14 | 0.48 | 8.50 | 2.50 |

**Table 1b.** Bulk geochemical data for the microbial mat (extraction residues after decalcification; modified after Blumenberg et al., 2015)

| Layers | $C_{org}$(%) | $N_{tot}$ (%) | $S_{tot}$ (%) | $C_{org}/N$ | $C_{org}/S$ |
|--------|--------------|---------------|---------------|-------------|-------------|
| 1 | 15.2 | 1.9 | 10.4 | 7.9 | 1.5 |
| 2 | 32.8 | 4.9 | 2.9 | 6.7 | 11.4 |
| 3 | 37.3 | 5.6 | 4.6 | 6.7 | 8.2 |
| 4 | 37.0 | 6.2 | 2.6 | 6.0 | 14.4 |
| 5 | 40.1 | 6.7 | 2.0 | 6.0 | 20.3 |
| 6 | 42.3 | 6.6 | 2.0 | 6.4 | 21.4 |

**Table 2a** Concentrations of sterols in the freely extractable lipids of the microbial mat layers. SD indicate mean value of standard deviation (μg/g dry mat; n.d. = not detected).

| Trivial names | Cholesterol | Cholestanol | Brassicasterol | Campesterol | Ergostanol | Stigmasterol | β-sitosterol | Stigmastanol | 4α-methylgorgosterol | 4α-methylgorgostanol |
|---|---|---|---|---|---|---|---|---|---|---|
| Compound<br><br>SD | Cholest-5-en-3β-ol | 5α-cholestan-3β-ol | 24-methylcholesta-5,22-dien-3β-ol | 24-methylcholest-5-en-3β-ol | 5α-24-methylcholestan-3β-ol | 24-ethylcholesta-5,22-dien-3β-ol | 24-ethylcholest-5-en-3β-ol | 5α-24-ethylcholestan-3β-ol | 22,23-methylene-4α,23,24-trimethylcholest-5-en-3β-ol | 22,23-methylene-4α,23,24-trimethylcholestan-3β-ol |
| Layers | 28.4% | 41.7% | 12.31% | 10.7% | 28.3% | 20.0% | 13.2% | 27.7% | 41.3% | 41.6% |
| 1 | 5.20 | 0.86 | 4.93 | 4.91 | 0.20 | 3.74 | 5.44 | 0.77 | 1.10 | 5.21 |
| 2 | 0.24 | 0.19 | n.d. | 0.34 | 0.06 | 0.43 | 0.77 | 0.19 | 0.36 | 1.55 |
| 3 | 0.01 | 0.04 | n.d. | 0.02 | 0.02 | 0.02 | 0.03 | 0.05 | 0.09 | 0.37 |
| 4 | 0.02 | <0.01 | n.d. | 0.02 | 0.01 | 0.05 | 0.02 | 0.02 | 0.12 | 0.40 |
| 5 | 0.03 | 0.02 | n.d. | 0.07 | 0.04 | 0.16 | 0.12 | 0.13 | 0.95 | 2.72 |
| 6 | 0.08 | <0.01 | n.d. | n.d. | n.d. | 0.08 | 0.02 | 0.01 | 0.10 | 0.73 |

**Table 2b** Concentrations of sterols in the carbonate-bound lipids of the microbial mat layers. SD indicate mean value of standard deviation (μg/g dry mat; n.d. = not detected; dashes indicate SD are not applicable).

| Trivial names | Cholesterol | Cholestanol | Brassicasterol | Campesterol | Ergostanol | Stigmasterol | β-sitosterol | Stigmastanol | 4α-methylgorgosterol | 4α-methylgorgostanol |
|---|---|---|---|---|---|---|---|---|---|---|
| Compound | Cholest-5-en-3β-ol | 5α-cholestan-3β-ol | 24-methylcholesta-5,22-dien-3β-ol | 24-methylcholest-5-en-3β-ol | 5α-24-methylcholestan-3β-ol | 24-ethylcholesta-5,22-dien-3β-ol | 24-ethylcholest-5-en-3β-ol | 5α-24-ethylcholestan-3β-ol | 22,23-methylene-4α,23,24-trimethylcholest-5-en-3β-ol | 22,23-methylene-4α,23,24-trimethylcholestan-3β-ol |
| Layers    SD | 24.0% | 47.6% | — | — | — | — | 24.8% | 40.2% | — | — |
| 1 | n.d. | n.d. | n.d. | n.d. | n.d. | n.d. | n.d. | n.d. | n.d. | n.d. |
| 2 | 0.02 | <0.01 | n.d. | 0.08 | <0.01 | 0.08 | 0.07 | 0.01 | n.d. | 0.34 |
| 3 | 0.04 | 0.05 | n.d. | 0.08 | 0.02 | 0.03 | 0.06 | 0.04 | n.d. | 0.40 |
| 4 | 0.11 | 0.07 | n.d. | 0.12 | 0.03 | 0.25 | 0.12 | 0.09 | n.d. | 0.45 |
| 5 | 0.02 | <0.01 | n.d. | n.d. | n.d. | 0.06 | 0.01 | <0.01 | n.d. | 0.22 |
| 6 | 0.02 | <0.01 | n.d. | 0.03 | <0.01 | 0.04 | 0.03 | 0.01 | n.d. | 0.59 |

**Table 3** Stanol/stenol ratios in the freely extractable lipids and carbonate-bound lipids for the microbial mat layers (n.d. = not determined, due to very low concentration of sterols).

| Layer | stanol/stenol ($\Delta^0/\Delta^5$) in Free lipids | | | | stanol/stenol ($\Delta^0/\Delta^5$) in carbonate-bound | | | |
| --- | --- | --- | --- | --- | --- | --- | --- | --- |
| | $C_{27}$ | $C_{28}$ | $C_{29}$ | $C_{31}$ | $C_{27}$ | $C_{28}$ | $C_{29}$ | $C_{31}$ |
| 1 | 0.17 | 0.04 | 0.14 | 4.73 | n.d. | n.d. | n.d. | n.d. |
| 2 | 0.77 | 0.18 | 0.24 | 4.26 | 0.34 | 0.05 | 0.19 | n.d. |
| 3 | 3.49 | 1.00 | 1.52 | 3.95 | 1.28 | 0.22 | 0.64 | n.d. |
| 4 | 0.38 | 0.42 | 1.24 | 3.31 | 0.62 | 0.27 | 0.74 | n.d. |
| 5 | 0.78 | 0.55 | 1.08 | 2.87 | n.d. | n.d. | n.d. | n.d. |
| 6 | n.d. | n.d. | n.d. | 7.35 | 0.11 | n.d. | 0.29 | n.d. |