# Peer review of "Sterol preservation in hypersaline microbial mats"

_Biogeosciences, 2019_

## Referee Comment (RC1) · Rienk Smittenberg (Referee) · 19 Jun 2019

Shen et al present a paper that is very similar in set-up as a paper of the same first authors published in 2017, where they analyse the sterol distribution in a microbial mat from a hypersaline lake on Christmas Island (Kirimati) - the main difference is that they now targeted another lake - which was previously investigated by the same group (Blumenberg et al 2015) who published hypy-GCMS results. This paper presents free sterols and some fatty acid data from the same microbial mat. The data is interesting and gives some additional insight in the lipid biomarker composition of these hypersaline mats, as a modern analog to ancient stromatolites.

I have some concerns that i like to be addressed before accepting as a final paper: * The use of TMCS in methanol to methylate fatty acids is rather uncommon. Please provide a reference where this method and its efficiency is described. * How were the 13C

contents of the sterols and FAs corrected for the added derivatizing groups? * I wonder if pyrolysis GC-MS is the best way to assess if steroids make it into the 'kerogen' (residue after extraction) fraction of recent material. I am not a py-GCMS expert but pyrolysis at 560'C is a rather high temperature where most organic molecules will break down to smaller pieces; any remaining intact compounds or larger fragments will be low in concentration. The fact that some hopanoids (fragments) may indicate that they are simply more abundant, while any remaining steroids could be below detection limit (i.e. below the background of the 213+215+217 trace). Absence of evidence is not the same as evidence of absence. Information about the reference kerogen of the Green River shale is lacking (e.g. sample amount, or relative amount of hopanes-steranes in Green River bitumen) so this comparison does not tell very much. More critically, Blumenberg (2015) could report hopane/sterane ratios for the same mat, which means steranes were present throughout – although they did appear to decrease with depth (at least compared to the hopanes). I would have the same critique to the already published paper about lake 22 (Shen 2018) when it concerns the use of py-GCMS to investigate the presence of steranes in the 'proto-kerogen'. * The paper cannot really be read without also consulting a more comprehensive hypy-GCMS- biomarker analysis published by Blumenberg (2015), who finds hopane/sterane ratios of 20-100 – thus no surprise there are no sterols found by py-GCMS while hopanes do show a trace.

* Why did the authors not measure (or present) free hopanoids? Or for that matter a more comprehensive biomarker study (i.e. a free extractable lipid version of the Blumenberg paper). This would give the paper much more body, constraining it to only steroids feels very limited. I strongly recommend expanding the paper in this way.

* The 14C dating of carbonates on a Coral atoll has a large risk of a reservoir effect (the coral is likely from the mid-Holocene sea level high stand thus several 1000 years old). Indeed a 14C age of just -239 years indicates a fraction modern of just over 1, i.e. a mixture of post-bomb atmospheric CO2 and an ancient source. The downcore increase in age does make sense, but one cannot assign any exact ages to the mat

material – for this, one needs to date plant macrofossils. I realise the results were published already by Blumenberg et al but they can only be interpreted as deeper = older. * When looking at the depth profiles of the sterols, I do not see a clear decrease with depth, except the large difference between the surface layer and layers below the phototrophic active part. Layer 5 and layer 2 have the same concentration per g TOC, and layer 3 and 6 the same expressed per g dry mat. * Comparison with lake 22 (Shen 2018): What is different between the two lakes is a halite-gypsum crust on top of lake 22 – which must impair oxygen flux to the upper layer. Yet, lake 22 shows considerably higher steroid concentrations than this lake 2. This may explain the absence of higher sterol abundances in the upper layer (i.e. absence of eukaryotic sterol-producing photosynthetic organisms in lake 22 but rather a 'fossil' signal starting already in layer 1 below the halite crust. The higher sterol conc. in lake 22 may simply be a higher contribution from terrestrial vegetation, but as the authors state it can also be a lower degradation because of ultrahigh salinity. Coprostanol found in lake 22 could be derived from the abundant land crabs on Kirimati, which live around the lakes and eat the local vegetation (and each other - personal observation in 2005). * I agree with the final conclusion that different microbial mats, like lake 22 and lake 2, generate different fossils records, because of their different limnic/environmental properties. * I also agree that the data confirm the hypothesis that microbial mats do not preserve original photosynthetic lipids from the upper very well, and that this signal is overprinted by heterotrophic organisms. However marine or lake sediments do have the same 'problem': organic matter degradation on heterotrophy within (anoxic) sediments, i.e. a diagenetic overprint. * That said, these are not really very new insights, the added value compared to the Blumenberg 2015 and Shen 2018 papers is marginal. * I do not agree with the conclusion that steroids are not preserved, in my opinion the authors have not used the right method to investigate this. Blumenberg (2015) found steranes after hypy.

---

## Referee Comment (RC2) · Gordon Love (Referee) · 25 Jun 2019

This study us a follow-up to two previous papers published on Kiribati hypersaline lake mat ecosystems, published by some of the same authors. While there is no over-whelming consensus from three studies, as to whether steroids are preferentially degraded over other lipid types due to taphonomic bias, the biomarker analyses are of good quality and the overall results are of general interest to organic geochemists and geobiologists.

I have three major comments on this work that I would like the authors to address:

1) Why do the authors assume that the microbial communities, and hence the lipid composition, should be constant over 1500 years of mat growth and burial?

They interpret differences in lipid composition to predominantly taphonomic factors

. . ..but this is based on an unsubstantiated assumption that eukaryotic contributions to the mat community were fairly constant over depositional history.

The depletion in sterols in deeper layers (in conflict with the results of Shen et al. that showed abundant steroid lipids in all mat layers from another Kirbibati lake) could just as easily indicate a changing mat biological community through time. With higher bacterial contributions to deeper versus shallow layers.

If a "mat-seal" bias was operational and a pervasive diagenetic mechanism for microbial mat remineralization and preservation then why do the results of Shen et al. contradict those found in this investigation?. It seems more likely that the differences in lipid composition represent temporal changes in mat community.

2) Given that Shen et al. could not find kerogen-bound steroids in a previous study using this same Py-GC-MS technique then it becomes suspicious that the pyrolysis method used in not optimized to detect bound steroids in degradation products from "young" mat sediments.

There should be no reason from first principles why bound steroids will not be found given that there is ample proto-kerogen in these mat sediments (given that sequestration will begin very early during diagenesis, see point 3 below)..

This might be due to high baselines in the ion chromatograms that they are searching for steranes and sterenes. Another reason is that Py-GC-MS will produce a complex mixture of unsaturated steroids and sterols bound within polar moieties after bond cleavage, with no good hydrogen donor in the system to cleave these out as steranes.

The authors could perhaps estimate what their detection limit is for detecting kerogen-bound steroids, giving the analytical complexities?

3) I refer the authors to a recent study by Lee et al. (2019) OG, which has only just appeared., in print that showed evidence for early diagenetic incorporation of biomarker lipids by covalent binding into benthic mat sediments from a salt pond Guerrero Negro,

Mexico . They used sequential chemolysis and HyPy degradation on extracted micro-bial mat sediments and found evidence for early diagenetic incorporation of a variety of linear, branched and polycyclic lipid skeletons into proto-kerogen on a timescale of only years to decades. The lipids includes bound hopanoids and bound steroids.

So, this supports the idea that HyPy is an effective method for trying to detect bound steroids in mat proto-kerogen due to the I) high sample capacity and ii) use of reducing conditions that yields appreciable steranes and sterene products. It further supports the idea as described in 2) that the Py-GC-MS method used in this investigation is maybe problematic for detecting immature bound steroids from proto-kerogen.

It is surprising since this group has their own HyPy equipment that they choose an online Py-GC-MS method to try and detect kerogen-bound steroids.

The amount of high mw and polar material in pyrolysates from "young" mat sediments. will be appreciable so it is best to choose a method that generates a substantial portion of bound steroids as hydrocarbon products. Even with HyPy, the "polar" fraction domi-nates the pyrolysate products so this problem will be even more acute with Py-GC-MS performed with an inert gas (rather than high pressure hydrogen and a catalyst as used in HyPy).

---

## Author Comment (AC1) · 8 Aug 2019

We sincerely thank referee #1 (Rienk Smittenberg) for the constructive comments, which helped us to improve our manuscript. Below we list all the points raised by the reviewer (given between quotes), followed by our replies.

1. – "The use of TMCS in methanol to methylate fatty acids is rather uncommon. Please provide a reference where this method and its efficiency is described."

REPLY: We now provide a reference describing the use and efficiency of TMCS in methanol for the methylation of fatty acids by Poerschmann and Carlson (2006) in the materials and methods section.

Planned changes in manuscript: We will cite this reference in the revised version (chapter 2.3).

2. – "How were the 13C contents of the sterols and FAs corrected for the added derivatizing group."

REPLY: We corrected 13C contents for the added derivatizing group following the method described by Goñi and Eglinton (1996). First, we analyzed Heneicosylic acid and Androstanol standards derivatized as methyl ester (ME-) and trimethylsilyl (TMS-) ester by GC-C-IRMS to obtain the carbon isotopic value for the derivatizing groups, and after measuring our samples, we corrected the carbon isotope values of our derivatized lipids according to the equations provided in that study. This will be indicated in the materials and methods section.

Planned changes in manuscript: We will specify the method and add the reference (Goñi and Eglinton, 1996) in the revised version (chapter 2.6).

3. – "I wonder if pyrolysis GC-MS is the best way to assess if steroids make it into the 'kerogen' fraction of recent material. I am not a Py-GC-MS expert but pyrolysis at 560°C is rather high temperature where most organic molecules will break down to smaller pieces; any remaining intact compounds or larger fragments will be low in concentration. The fact that some hopanoids (fragments) may indicate that they are simply more abundant, while any remaining steroids could be below detection limit (i.e. below the background of the 213+215+217 trace). Absence of evidence is not the same as evidence of absence...,"

REPLY: As demonstrated by for example Gelin et al. (1996), steroids are thermally stable at such high temperatures (610°C in Gelin et al., 1996) and Py-GC-MS is a suitable method to demonstrate their presence in immature kerogens. Another example demonstrating the value of Py-GC-MS for our objectives is the study by Kruge and Permanyer (2004), who applied Py-GC-MS at 600°C for evaluating steranes/sterenes as tracers for organic contamination in recent sediments. Finally, the results obtained from our reference material can be taken as proof that the applied method is suitable for the analysis of steroids bound to kerogen. These points will be indicated in the results and

discussion section.

We agree that "Absence of evidence is not the same as evidence of absence". In this regard, we will reword the discussion chapter and clarify that the kerogen-bound steroids are below our (individual) detection limit for Py-GC-MS. Regarding the definition of our detection limit, please see author's response below (5.).

Planned changes in manuscript: We will reword and add these points in the revised version (chapter 3.4 and 4.2).

4. – "...Information about the reference kerogen of the Green River shale is lacking (e.g. sample amount, or relative amount of hopanes-steranes in Green River bitumen) so this comparison does not tell very much."

REPLY: Our intention when displaying the reference run was to demonstrate the ability of the system to detect the compounds in question, and to indicate the exact retention times in the chromatograms. For that purpose we used about 0.5 mg of kerogen purified from the Eocene Green River oil shale (Eastern Utah, White River Mine, BLM Oil Shale Research, Development, and Demonstration Lease UTU-84087).

Planned changes in manuscript: We will add detailed information about the reference material to the revised version (chapter 3.4; supplementary information).

5. – "More critically, Blumenberg (2015) could report hopane/sterane ratios for the same mat, which means steranes were present throughout – although they did appear to decrease with depth (at least compared to the hopanes). I would have the same critique to the already published paper about Lake 22 (Shen 2018) when it concerns the use of Py-GC-MS to investigate the presence of steranes in the 'proto-kerogen.'"

REPLY: It is generally difficult to provide sound quantitative data with Py-GC-MS. An estimate for the detection limit of our system can be derived from our analysis of the samples together with an internal standard (n-eicosane D42, 120 ng) that was routinely added to check the performance of the chromatographic system. When comparing our

steroid peaks to this standard (and neglecting slightly different response factors), the amount of analyte should be about 1 ng (absolute amount) to obtain a reliable mass spectrum that allows for identification. We admit that this is certainly higher than the detection limit of the HyPy-approach (Blumenberg et al. 2015), that includes chromatographic isolation, catalytic hydrogenation and concentration of the target analytes. This might partly explain the lack of steroid signals in our chromatograms (see also 6.).

To clarify how steroids can be detected by our Py-GC-MS system, we have added below a screenshot (Figure 1) showing Py-GC-MS chromatograms (TICs) of the original microbial mat (Layer 2; upper chromatogram), and its extraction residue ('kerogen'; lower chromatogram). Sterenes and hopenes are abundant in the original mat sample, but their concentrations are below detection limit in the kerogen. Mass spectra for some compounds from the upper chromatogram are also given, representing a C27 sterene (1A), a C27 hopene (2A), and a C28 sterene (3A). Compound 3A is just identifiable from the TIC (though some coelution is evident in the mass spectrum) and it was therefore used to define our (conservative) detection limit of ∼1 ng (absolute amount) for kerogen-bound steroids.

Planned changes in manuscript: We will detail our detection limit, and clarify that our pyrolysis results cannot exclude the presence of small amounts of steroids in the kerogen fractions (abstract; chapter 3.4 and 4.2).

6. – "The paper cannot really be read without also consulting a more comprehensive hypy-GC-MS- biomarker analysis published by Blumenberg (2015), who finds hopane/sterane ratios of 20-100- thus no surprise there are no sterols found by Py-GC-MS while hopanes do show a trace."

REPLY: As detailed above (5.), we agree that small amounts of steroids might not be detected by Py-GC-MS due to our detection limit.

Planned changes in manuscript: We will provide an improved discussion including a more detailed comparison with the results from Blumenberg et al. (2015), and a

discussion of the detection limit (chapter 4.2).

7. – "Why did the authors not measure (or present) free hopanoids: Or for that matter a more comprehensive biomarker study (i.e. a free extractable lipid version of the Blumenberg paper). This would give the paper much more body, constraining it to only steroids feels very limited. I strongly recommend expanding the paper in this way."

REPLY: We agree that hopanoids may provide additional information, but our study was aimed at the fate of steroids. Whereas our GC-MS based approach is able to assess the whole range of steroids, it can only reveal some diagenetic products of the original polyfunctionalized hopanoids, e.g. hopanes/hopenes, and hopanols. Providing a comprehensive picture on the hopanoids would also require the additional analysis of polyfunctionalized hopanoids, and thus, the use of further techniques such as LC-MS, which we feel is beyond the scope of this study.

8. – "The 14C dating of carbonates on a coral atoll has a large risk of a reservoir effect (the coral is likely from the mid-Holocene sea level high stand thus several 1000 years old). Indeed a 14C age of just -239 years indicates a fraction modern of just over 1, i.e. a mixture of post-bomb atmospheric $CO_2$ and an ancient source. The downcore increase in age does make sense, but one cannot assign any exact ages to the mat material - for this, one needs to date plant macrofossils. I realize the results were published already by Blumenberg et al but they can only be interpreted as deeper=older."

REPLY: Since we consider the exact ages of these layers irrelevant for our purposes, we will follow the advice of the reviewer and not expand on the 14C data published by Blumenberg et al. (2015).

Planned changes in manuscript: We will remove the detailed age information (Figure 1 and chapter 3.1).

9. – "When looking at the depth profiles of the sterols, I do not see a clear decrease with depth, except the large difference between the surface layer and layers below the

phototrophic active part. Layer 5 and layer 2 have the same concentration per g TOC, and layer 3 and 6 the same expressed per g dry mat."

REPLY: In the manuscript, we refer to the same "large difference" between the surface and the deeper layers, as noted by the reviewer (e.g. P7, line 7-11). We did not mean that there was a gradual decrease with depth.

Planned changes in manuscript: The respective text passages will be reworded to clarify this (chapter 3.3).

10. – "Comparison with Lake 22 (Shen 2018): What is different between the two lakes is a halite-gypsum crust on top of Lake 22 – which must impair oxygen flux to the upper layer. Yet, Lake 22 shows considerably higher steroid concentrations than this Lake 2. This may explain the absence of higher sterol abundances in the upper layer (i.e. absence of eukaryotic sterol-producing photosynthetic organisms in Lake 22 but rather a 'fossil' signal starting already in layer 1 below the halite crust. The higher sterol conc. in Lake 22 may simply be a higher contribution from terrestrial vegetation, but as the authors state it can also be a lower degradation because of ultrahigh salinity. Coprostanol found in Lake 22 could be derived from the abundant land crabs on Kiritimati, which live around the lakes and eat the local vegetation (and each other - personal observation in 2005)."

REPLY: We agree that the halite-gypsum crust on top of the Lake 22 mat may have impaired oxygen flux to the upper layer, thereby reducing the production of sterols in the upper part of the Lake 22 mat. On the other hand, we do not consider the higher sterol concentrations in Lake 22 as being caused by a higher terrestrial input. This is demonstrated by the distributions of sterol pseudohomologues (high in C27) as well as the similar sterol concentrations in the top layers of both mats (102 $\mu$g/g Corg range, see chapter 4.3, P10, line 26-27). As discussed in the manuscript, we consider hypersalinity, combined with periods of subaerial exposure, as more important factors on the degradation rates, and regard this a more likely explanation why steroids showed

no overall systematic decrease throughout the mat profile in Lake 22.

Planned changes in manuscript: A discussion on the potential effect of a halite-gypsum crust on top of Lake 22 mat will be added to the revised version (chapter 4.3).

11. – "I also agree that the data confirm the hypothesis that microbial mats do not preserve original photosynthetic lipids from the upper very well, and that this signal is overprinted by heterotrophic organisms. However, marine or lake sediments do have the same' problem': organic matter degradation on heterotrophy within (anoxic) sediments, i.e. a diagenetic overprint. That said, these are not really very new insights, the added value compared to the Blumenberg 2015 and Shen 2018 papers is marginal."

REPLY: In the current work, we investigated a microbial mat from a lake with completely different environmental conditions as compared to the 2018 publication with respect to salinity and water depth. In addition, the microbial mat studied in 2018 probably experienced periods of subaerial exposure, which is not observed for the currently studied microbial mat. Further, unlike it has been done in the 2018 paper and in Blumenberg et al. (2015), we explicitly analyzed steroid distributions in freely extractable vs. carbonate-bound lipid fractions to test whether calcification within the microbial mat may function as a preservation mechanism for these biomarkers.

Planned changes in manuscript: We will reword and put more emphasis on the novel insights that our new work provided as compared to Blumenberg et al. (2015) and Shen et al. (2018) (abstract, chapter 4.3).

12. – "I do not agree with the conclusion that steroids are not preserved, in my opinion the authors have not used the right method to investigate this. Blumenberg (2015) found steranes after hypy."

REPLY: The reviewer is correct in that it cannot be concluded that steroids are not preserved in microbial mats, but we did not claim this in our manuscript. For the Lake 2 mat we demonstrate how sterols experienced major degradation that suppressed the

primary ecological signal. In contrast, sterols in the Lake 22 mat (Shen et al., 2018) experienced major microbial transformation which largely preserved their molecular integrity. Taken together, these findings highlight that sterols may have contrasting preservation pathways in microbial mats, and that preservation may be much better in mats experiencing higher salinities and/or more desiccated conditions. This is certainly a relevant and novel insight revealed by this study.

Planned changes in manuscript: We will put more emphasis on the novel findings and sharpen the respective text passages in abstract, discussion, and conclusions sections (chapters 4.2, 4.3 and 5).

References cited in the reply:

Blumenberg M., Thiel V. and Reitner J. (2015). Organic matter preservation in the carbonate matrix of a recent microbial mat – Is there a 'mat seal effect'? Organic Geochemistry 87, 25–34.

Gelin F., Sinninghe Damsté J. S., Harrison W. N., Reiss C., Maxwell J. R. and De Leeuw J. W. (1996) Variations in origin and composition of kerogen constituents as revealed by analytical pyrolysis of immature kerogens before and after desulphnrization. Organic Geochemistry 24, 705–714.

Goñi M. A. and Eglinton T. I. (1996) Stable carbon isotopic analyses of lignin-derived CuO oxidation products by isotope ratio monitoring-gas chromatography-mass spectrometry (irm-GC-MS). Organic Geochemistry 24, 601–615.

Kruge M. A. and Permanyer A. (2004) Application of pyrolysis-GC/MS for rapid assessment of organic contamination in sediments from Barcelona harbor. Organic Geochemistry 35, 1395–1408.

Poerschmann J. and Carlson R. (2006) New fractionation scheme for lipid classes based on "in-cell fractionation" using sequential pressurized liquid extraction. Journal of chromatography. A 1127, 18–25.

Shen Y., Thiel V., Duda J.-P. and Reitner J. (2018). Tracing the fate of steroids through a hypersaline microbial mat (Kiritimati, Kiribati/Central Pacific). Geobiology 16, 307–318.

Please also note the supplement to this comment:
https://www.biogeosciences-discuss.net/bg-2019-124/bg-2019-124-AC1-supplement.pdf

———————————————————

[Figure]

[Figure]

Fig. 1.

---

## Author Response (AR1)

**GEORG—AUGUST-UNIVERSITY OF GOETTINGEN**

**Centre of Geoscience**
**Department of Geobiology**
**Yan Shen**

[Figure]

Goldschmidtstr. 3
37077 Göttingen
Germany
Phone: +49(0)551-39 7955
Fax: +49(0)551-39 7918
E-mail: yshen@gwdg.de
http://www.
geobiologie.uni-goettingen.de

Univ.Göttingen▪ GZG▪Abt.Geobiologie▪Goldschmidtstr.3▪37077 Göttingen▪Germany

**Associate Editor** *Biogeosciences*
Dr. Marcel van der Meer

Göttingen, 20.11.2019

**Submission of the revised manuscript ("*Sterol preservation in hypersaline microbial mats*"; bg-2019-214)**

Dear Dr. Marcel van der Meer,

also on behalf of my co-authors, I would like to thank you, Dr. Rienk Smittenberg and Dr. Gordon Love for the thorough and constructive comments on the manuscript. All of the comments have been carefully considered, and the suggested corrections have been added into the revised version.

In order to help you to track our changes and modifications, we uploaded the following documents:

1. Reply to your comments
2. Replies to the comments of all reviewers (RC1-2 from Nov 20th)
3. Tracked changes version of the manuscript (main text)
4. The new version of the manuscript + all figures, tables and supplementary documents

We trust that the revised manuscript will meet the requirements and demands of the reviewers. I would appreciate a reply at your earliest convenience.

Thank you for your consideration.

Yours Sincerely,

Yan Shen
**Responses from the authors to the comments of the subject editor (Dr. Marcel van der Meer) on manuscript bg-2019-124**

We sincerely thank the associated editor (Dr. Marcel van der Meer) for the helpful and constructive comments that improved our manuscript.

Comment from the editor: "Both reviewers, and myself, have mentioned that this paper seems to be a continuation of already published work with quite some overlap in the author lists as well. I have mentioned before that you have to make it very clear why this manuscript is able to stand on its own and at least one reviewer made a similar remark, so make that absolutely clear in the rebuttal and revised version."

Author's response: The study has been published in Geobiology 2018, at first glance, is similar to the current one. However, in that study we analyzed microbial mat samples from another lake, and there are significant differences in the focus and techniques that we used, and also the outcome of both studies is remarkably different. Furthermore, in the study of Blumenberg et al. (2015), the concentrations of individual steroid compounds have not been provided for our mat in this work.

For the work now under review for Biogeosciences, we explicitly discuss steroid distributions in freely extractable vs. carbonate-bound lipid fractions as well as in the decalcified extraction residues. We also test if calcification within the microbial mat may function as a preservation mechanism for these biomarkers. These points were apparently not included in the 2018 publication. In addition to steroids, new data such as hopanoids and fatty acids requested by reviewer #1 are now included and discussed in the revised version (chapter 3.3, 3.4, 3.5 and 4.2).

Furthermore, in our current study, we observed how total sterol concentrations decreased immediately below the mat surface, thus supporting the "mat-seal effect" hypothesis. This is remarkably different from the 2018 publication, where there was no evidence for such a mat-seal effect. In the discussion, we considered potential mechanisms which might have caused the distinct behavior of sterols and also tried to examine general trends in the preservation of sterols in hypersaline microbial mat systems.

Therefore, this manuscript provides new data and novel insights into the preservation pathways of steroid biomarkers in microbial mats. We trust that these results, together with the analysis of a microbial mat from a different lake, and the different focus and techniques we used, justifies the publication of this work as a stand-alone paper in Biogeosciences.

Comment from the editor: "Both reviewers commented on the detection limit of your method and how this might relate to your detection of sterols or absence thereof. Make sure you address this issue and how this might affect your results and discussion. I am not really happy with your reply to the comment about HyPy versus your Py results of Dr. Gordon Love, please try to come up with a better answer."

Author's response: Indeed, producing reproducible quantitative data with Py-GC-MS is difficult. We further agree with the reviewer that the detection limit of our Py-GC-MS may be higher than the

HyPy-approach and have rephrased the discussion in the revised version. HyPy includes catalytic hydrogenation, chromatographic isolation and concentration of the target analytes, together resulting in less interferences and a relatively higher sensitivity. An estimate for the detection limit of our system can be derived from our analysis of the samples together with an internal standard (*n*-eicosane D42, 120 ng) that was routinely added to check the performance of the chromatographic system. When comparing our steroid peaks to this standard (and neglecting slightly different response factors), the absolute amount of analyte should be ~1 ng to obtain a reliable signal enabling identification via the mass spectrum. The limited sensitivity of our Py-GC-MS system may have resulted in non-detection of small amount of steroids, which are obviously still detectable by HyPy (Blumenberg et al., 2015). Nonetheless, both techniques clearly show that hopanoids are more abundant by far than steroids in the Lake 2 mat. We newly discussed this aspect and also included the findings made by Lee et al. (2019) into the discussion. We do not agree, however, with the reviewer's feeling that our method is generally not useful for our purpose; many studies have demonstrated the capability of Py-GC-MS for analysing steroid moieties in macromolecular OM (e. g. Gelin et al., 1996; Kruge and Permanyer, 2004). Furthermore ,the ability of our instrument for steroid identification was also positively checked using non-extracted original mats and the Eocene Green River Shale as reference materials (see also replies 3, 5 and 6 to reviewer #1).

Comment from the editor: "I do have one other remark concerning microbial mats, degradation and time that I think connects to some of the remarks made by both Dr. Rienk Smittenberg and Dr. Gordon Love. Different from a typical sedimentary setting microbial mats do not simply get buried by material falling on top and therefore getting older with depth. As I understand it, you have photoautotrophic organisms at the top generating biomass and heterotrophic organisms "below" (usually the separations are not so definitive, but to keep it simple). The mats typically grow upwards on top of dead and reworked organic matter from the mat itself. Potentially even similar to say year rings in trees, so new mats, consisting of two layers grown on top of the compacted layers of last year/growing season. The carbonate layer could even be the results of microbial activity, separating layers. This would fit with the observation that age does not increase regularly with depth or that concentration does not decrease linearly with depth, but in steps. High in the fresh photoautotroph layer, lower in the heterotroph layer increasing again into the older photoautotroph layer and lowest in the old heterotroph layer, for instance. This also means that community composition changes with depth, even if it doesn't really change with time (the top layer will always be photoautotrophic). How do your results relate to mat development, is there any information how this mat develops over time?"

Author's response: We agree with the considerations that microbial mats evolve internally through a continuous reworking of organic matter from the own mat, and that microbial activity can be linked to mineral precipitation within the mat. In accordance, we have rewritten and clarified the description of the growth phases of the mat (section 3.1), and we have included (in sections 4.2.1 and 4.2.2) discussion about how some biomarker trends (i.e. stanol/stenol ratios for $C_{27}$-$C_{29}$ pairs and hopanoids abundance) indicate differences in microbial activity and microbial transformation of stenols in the different growth phases of the mat. Especially interesting, and in agreement with the editor's comments, is the case of the continuous mineral crust of layer 3, where intense microbial activity is recorded by the biomarkers. Concerning the timing of mat development, and in response to comments of Dr. Rienk Smittenberg, we have considered that the exact ages of the individual mat layers are actually not relevant for the purpose of this work and therefore, we have omitted the $^{14}$C data and any interpretation or discussion based on them, retaining only the description of the growth phases of the mat, which is more relevant, as indicated by the editor.
**Responses from the authors to the comments of Referee #1 (Dr. Rienk Smittenberg) on manuscript bg-2019-124**

yshen@gwdg.de

We are very grateful to referee #1 (Dr. Rienk Smittenberg) for his valuable and thoughtful comments that helped us to improve our manuscript. Below we list all the points raised by the reviewer (given between quotes), followed by our replies.

1. – "The use of TMCS in methanol to methylate fatty acids is rather uncommon. Please provide a reference where this method and its efficiency are described."

**REPLY**: We now provide a reference describing the use and efficiency of TMCS in methanol for the methylation of fatty acids by Poerschmann and Carlson (2006) in the materials and methods section.

**Changes in manuscript**: We have cited this reference in the revised version (chapter 2.3).

2. – "How were the $^{13}$C contents of the sterols and FAs corrected for the added derivatizing group."

**REPLY**: We corrected $^{13}$C contents for the added derivatizing group following the method described by Goñi and Eglinton (1996). First, we used GC-C-IRMS to analyze *n*-Heneicosanoic acid and Androstanol standards as non-derivatized lipids. Second, they are derivatized as methyl ester (ME-) and trimethylsilyl (TMS-) ethers to obtain the carbon isotopic values of the derivatizing groups. After measuring our samples, we calibrated the carbon isotope values of our derivatized lipids according to the equations provided in that study. This has been indicated in the materials and methods section.

**Changes in manuscript:** We now have specified the method and added the reference (Goñi and Eglinton, 1996) in the revised version (chapter 2.6).

3. – "I wonder if pyrolysis GC-MS is the best way to assess if steroids make it into the 'kerogen' fraction of recent material. I am not a Py-GC-MS expert but pyrolysis at 560°C is rather high temperature where most organic molecules will break down to smaller pieces; any remaining intact compounds or larger fragments will be low in concentration. The fact that some hopanoids (fragments) may indicate that they are simply more abundant, while any remaining steroids could be below detection limit (i.e. below the background of the 213+215+217 trace). Absence of evidence is not the same as evidence of absence…,"

**REPLY**: As demonstrated by Gelin et al. (1996), steroids are thermally stable at such high temperatures at 610°C and Py-GC-MS is a suitable method to demonstrate their presence in immature kerogens. Another example proving the value of Py-GC-MS for our objectives is the study by Kruge and Permanyer (2004), who applied Py-GC-MS at 600°C for evaluating steranes/sterenes as tracers for organic contamination in recent sediments. Finally, the results obtained from our Eocene Green River Shale reference material can be taken as proof that the applied method is appropriate for the analysis of kerogen-bound steroids. These points have been indicated in the results and discussion section.

We agree that *"Absence of evidence is not the same as evidence of absence"*. In this regard, we have reworded the discussion chapter and clarified that the kerogen-bound steroids are below the detection

limit for our Py-GC-MS. Regarding the definition of our detection limit, please see author's response below (5.).

**Changes in manuscript**: We have reworded and add these points in the revised version (chapter 2.5, 3.5 and 4.2.3).

4. – "…Information about the reference kerogen of the Green River shale is lacking (e.g. sample amount, or relative amount of hopanes-steranes in Green River bitumen) so this comparison does not tell very much."

**REPLY**: Our intention when displaying the reference run was to prove the capability of the system to detect the compounds in question, and to indicate the exact retention times in the chromatograms. For that purpose we used about 0.5 mg of kerogen purified from the Eocene Green River oil shale (Eastern Utah, White River Mine, BLM Oil Shale Research, Development, and Demonstration Lease UTU-84087).

**Changes in manuscript**: We have now included the detailed information about the reference material to the revised version (chapter 2.5; supplementary information).

5. – "More critically, Blumenberg (2015) could report hopane/sterane ratios for the same mat, which means steranes were present throughout – although they did appear to decrease with depth (at least compared to the hopanes). I would have the same critique to the already published paper about Lake 22 (Shen 2018) when it concerns the use of Py-GC-MS to investigate the presence of steranes in the 'proto-kerogen."

**REPLY**: The hopane/sterane ratios reported by Blumenberg et al. (2015) showed that hopane concentrations are more than 20 times higher than steranes in the mat that we also analyzed for this work. Moreover, Blumenberg et al. (2015) has not provided abundances of individual steroid compounds, making it difficult to directly compare their data with our results.

In addition, providing sound quantitative data with Py-GC-MS is generally difficult. An estimate for the detection limit of our system can be derived from our analysis of the samples together with an internal standard (*n*-eicosane D42, 120 ng) that was routinely added to check the performance of the chromatographic system. When comparing our steroid peaks to this standard (and neglecting slightly different response factors), the amount of analyte should be about 1 ng (absolute amount) to obtain a reliable mass spectrum that allows for identification. This might be higher than the detection limit of the HyPy-approach (Blumenberg et al. 2015), that includes chromatographic isolation, catalytic hydrogenation and concentration of the target analytes. These fundamental characteristic of HyPy might partly explain the lack of steroid signals in our chromatograms (see also 6.).

To clarify how steroids can be detected by our Py-GC-MS system, we have added a screenshot below showing Py-GC-MS chromatograms (TICs) of the original microbial mat (Layer 2; upper chromatogram), and its extraction residue ('kerogen'; lower chromatogram). Sterenes and hopenes are abundant in the original mat sample, but their concentrations are below detection limit in the kerogen. Mass spectra for some compounds from the upper chromatogram are also given, representing a $C_{27}$ sterene (1A), a $C_{27}$ hopene (2A), and a $C_{28}$ sterene (3A). Compound 3A is just identifiable from the TIC (though some coelution is evident in the mass spectrum) and it was therefore used to define our (conservative) detection limit of ~1 ng (absolute amount) for kerogen-bound steroids.

[Figure]

**Changes in manuscript**: We have provided the details of our detection limit, and clarified that our pyrolysis results cannot exclude the presence of small amounts of steroids in the kerogen fractions (chapter 3.5 and 4.2.3).

6. – "The paper cannot really be read without also consulting a more comprehensive hypy-GC-MS-biomarker analysis published by Blumenberg (2015), who finds hopane/sterane ratios of 20-100- thus no surprise there are no sterols found by Py-GC-MS while hopanes do show a trace."

**REPLY**: As detailed above (5.), we agree that small amounts of steroids might not be detected by Py-GC-MS due to our detection limit.

**Changes in manuscript**: We have provided an improved discussion including a more detailed comparison with the results from Blumenberg et al. (2015), and a discussion of the detection limit (chapter 4.2.3).

7. – "Why did the authors not measure (or present) free hopanoids: Or for that matter a more comprehensive biomarker study (i.e. a free extractable lipid version of the Blumenberg paper). This would give the paper much more body, constraining it to only steroids feels very limited. I strongly recommend expanding the paper in this way."

**REPLY**: We agree that hopanoids could provide additional information. We have therefore further conducted lab analysis of hopanoids to make comparison with steroids. In addition to hopanoids, we now also add fatty acids data to obtain a complementary picture of individual lipids and to a better understanding of preservation pathways of biomarker study within the mat studied. In order to address the taphonomy of steroids, hopanoids and fatty acids, we have re-structured the manuscript and discussed these three compounds classes in the different lipid fractions, respectively, including freely extractable lipids, carbonate-bound lipids as well as decalcified extraction residues.

**Changes in manuscript**: We have included detailed information about hopanoids and fatty acids data in the revised version (chapter 3.3, 3.4, 3.5 and 4.2).

8. – "The $^{14}$C dating of carbonates on a coral atoll has a large risk of a reservoir effect (the coral is likely from the mid-Holocene sea level high stand thus several 1000 years old). Indeed a $^{14}$C age of just -239 years indicates a fraction modern of just over 1, i.e. a mixture of post-bomb atmospheric $CO_2$ and an ancient source. The downcore increase in age does make sense, but one cannot assign any exact ages to the mat material - for this, one needs to date plant macrofossils. I realize the results were published already by Blumenberg et al but they can only be interpreted as deeper=older."

**REPLY**: Since we consider the exact ages of these layers irrelevant for our purposes, we have followed the advice of the reviewer and not expanded on the $^{14}$C data published by Blumenberg et al. (2015).

**Changes in manuscript**: We have removed the detailed age information (Figure 1 and chapter 3.1).

9. – "When looking at the depth profiles of the sterols, I do not see a clear decrease with depth, except the large difference between the surface layer and layers below the phototrophic active part. Layer 5 and layer 2 have the same concentration per g TOC, and layer 3 and 6 the same expressed per g dry mat."

**REPLY**: In the manuscript, we refer to the same "large difference" between the surface and the deeper layers, as noted by the reviewer. We did not mean that there was a gradual decrease with depth.

**Changes in manuscript**: The respective text passages have been reworded to clarify this (chapter 3.3.1).

10. – "Comparison with Lake 22 (Shen 2018): What is different between the two lakes is a halite-gypsum crust on top of Lake 22 – which must impair oxygen flux to the upper layer. Yet, Lake 22 shows considerably higher steroid concentrations than this Lake 2. This may explain the absence of higher sterol abundances in the upper layer (i.e. absence of eukaryotic sterol-producing photosynthetic organisms in Lake 22 but rather a 'fossil' signal starting already in layer 1 below the halite crust. The higher sterol conc. in Lake 22 may simply be a higher contribution from terrestrial vegetation, but as the authors state it can also be a lower degradation because of ultrahigh salinity. Coprostanol found in Lake 22 could be derived from the abundant land crabs on Kiritimati, which live around the lakes and eat the local vegetation (and each other - personal observation in 2005)."

**REPLY**: We agree that the halite-gypsum crust on top of the Lake 22 mat may have impaired oxygen flux to the upper layer, thereby reducing the production of sterols in the upper part of the Lake 22 mat. However, it is not related to the current study, when explaining the potential reasons for the distinct degradation patterns of steroids between Lake 2 and 22 mats. On the other hand, we do not consider the higher sterol concentrations in Lake 22 as being caused by a higher terrestrial input. This is demonstrated by the distributions of sterol pseudohomologues (high in $C_{27}$) as well as the similar sterol concentrations in the top layers of both mats ($10^2$ µg/g $C_{org}$ range, see chapter 4.3). As discussed in the manuscript, we consider hypersalinity, combined with periods of subaerial exposure, as more

important factors on the degradation rates, and regard this a more likely explanation why steroids showed no overall systematic decrease throughout the mat profile in Lake 22.

11. – "I also agree that the data confirm the hypothesis that microbial mats do not preserve original photosynthetic lipids from the upper very well, and that this signal is overprinted by heterotrophic organisms. However, marine or lake sediments do have the same' problem': organic matter degradation on heterotrophy within (anoxic) sediments, i.e. a diagenetic overprint. That said, these are not really very new insights, the added value compared to the Blumenberg 2015 and Shen 2018 papers is marginal."

**REPLY**: In the current work, we investigated a microbial mat from a lake with completely different environmental conditions (e.g., salinity and water depth) as compared to the 2018 publication. In addition, the microbial mat studied in 2018 probably experienced periods of subaerial exposure, which is not observed for the currently studied microbial mat. Further, unlike it has been done in the 2018 paper and in Blumenberg et al. (2015), we explicitly analyzed individual lipid distributions, including steroids, hopanoids and fatty acids, in freely extractable and carbonate-bound lipid fractions as well as in decalcified extraction residues. Moreover, we also test whether calcification within the microbial mat may function as a preservation mechanism for eukaryote-derived steroids.

**Changes in manuscript**: We have reworded and put more emphasis on the novel insights that our new work provided as compared to Blumenberg et al. (2015) and Shen et al. (2018) (chapter 3.3, 3.4, 3.5, 4.2 and 4.3). Please also see reply to the first comment from associated editor.

12. – "I do not agree with the conclusion that steroids are not preserved, in my opinion the authors have not used the right method to investigate this. Blumenberg (2015) found steranes after hypy."

**REPLY**: The reviewer is correct in that it cannot be concluded that steroids are not preserved in microbial mats, but we did not claim this in our manuscript. For the Lake 2 mat we demonstrate how sterols experienced major *degradation* that largely eliminated the primary eukaryotic signal. In contrast, sterols in the Lake 22 mat (Shen et al., 2018) experienced major microbial *transformation* which greatly preserved their molecular integrity. On the other hand, Blumenberg et al. (2015) has not provided abundances of individual steroid compounds, making it difficult to compare their data with our results.

Our findings highlight that sterols may have contrasting preservation pathways in microbial mats, and that preservation may be much better in mats experiencing higher salinities and/or more desiccated conditions. This is certainly a relevant and novel insight revealed by this study.

**Changes in manuscript**: We have emphasized on the novel findings and sharpen the respective text passages in abstract, discussion, and conclusions sections (chapters 4.2, 4.3 and 5).

References cited in the reply:

Blumenberg M., Thiel V. and Reitner J. (2015). Organic matter preservation in the carbonate matrix of a recent microbial mat – Is there a 'mat seal effect'? Organic Geochemistry 87, 25–34.

Gelin F., Sinninghe Damsté J. S., Harrison W. N., Reiss C., Maxwell J. R. and De Leeuw J. W. (1996) Variations in origin and composition of kerogen constituents as revealed by

analytical pyrolysis of immature kerogens before and after desulphnrization. Organic Geochemistry 24, 705–714.

Goñi M. A. and Eglinton T. I. (1996) Stable carbon isotopic analyses of lignin-derived CuO oxidation products by isotope ratio monitoring-gas chromatography-mass spectrometry (irm-GC-MS). Organic Geochemistry 24, 601–615.

Kruge M. A. and Permanyer A. (2004) Application of pyrolysis-GC/MS for rapid assessment of organic contamination in sediments from Barcelona harbor. Organic Geochemistry 35, 1395–1408.

Poerschmann J. and Carlson R. (2006) New fractionation scheme for lipid classes based on "in-cell fractionation" using sequential pressurized liquid extraction. Journal of chromatography. A 1127, 18–25.

Shen Y., Thiel V., Duda J.-P. and Reitner J. (2018). Tracing the fate of steroids through a hypersaline microbial mat (Kiritimati, Kiribati/Central Pacific). Geobiology 16, 307–318.
**Responses from the authors to the comments of Referee #2 (Dr. Gordon Love) on manuscript bg-2019-124**

yshen@gwdg.de

We are very grateful to referee #2 (Dr. Gordon Love) for his helpful and constructive comments on our manuscript. Below we list all the points raised by the reviewer (given between quotes), followed by our replies.

1. – "Why do the authors assume that the microbial communities, and hence the lipid composition, should be constant over 1500 years of mat growth and burial? They interpret differences in lipid composition to predominantly taphonomic factors… but this is based on an unsubstantiated assumption that eukaryotic contributions to the mat community were fairly constant over depositional history.

The depletion in sterols in deeper layers (in conflict with the results of Shen et al. that showed abundant steroid lipids in all mat layers from another Kiribati lake) could just as easily indicate a changing mat biological community through time. With higher bacterial contributions to deeper versus shallow layers.

If a "mat-seal" bias was operational and a pervasive diagenetic mechanism for microbial mat remineralization and preservation then why do the results of Shen et al. contradict those found in this investigation? It seems more likely that the differences in lipid composition represent temporal changes in mat community."

**REPLY**: Although the possibility of a change in microbial communities cannot be excluded, we observed no indications that eukaryotic contributions would have drastically changed over time. So far, the studies on microbial mats from Kiritimati have indicated eukaryotic contributions (Bühring et al., 2009; Blumenberg et al., 2015; Shen et al., 2018), and there are no indications why there should be no eukaryotic inputs at the time when the deeper parts of the mat had been formed. Further, there seem to be no major changes in the texture of the carbonate phases of the mat (except the thin mineral crust representing layer 3), which would suggest major changes in the microbial community. As already discussed, we suggest that the salinity levels and periods of subaerial exposure could have caused the differences observed between the microbial mats from Lake 2 and Lake 22.

**Changes in manuscript**: We have included a brief discussion why we consider a decrease in eukaryotic input over time to be a less likely cause for the fluctuations in steroid distributions as compared to environmental factors (chapter 4.2.1).

2. – "Given that Shen et al. could not find kerogen-bound steroids in a previous study using this same Py-GC-MS technique then it becomes suspicious that the pyrolysis method used in not optimized to detect bound steroids in degradation products from "young" mat sediments.

There should be no reason from first principles why bound steroids will not be found given that there is ample proto-kerogen in these mat sediments (given that sequestration will begin very early during diagenesis, see point 3 below).

This might be due to high baselines in the ion chromatograms that they are searching for steranes and sterenes. Another reason is that Py-GC-MS will produce a complex mixture of unsaturated steroids and sterols bound within polar moieties after bond cleavage, with no good hydrogen donor in the system to cleave these out as steranes.

The authors could perhaps estimate what their detection limit is for detecting kerogen-bound steroids, giving the analytical complexities?"

**REPLY**: Please see author's response to referee #1 (5.).

**Changes in manuscript**: Please see author's response to referee #1 (5.).

3. – "I refer the authors to a recent study by Lee et al. (2019) OG, which has only just appeared., in print that showed evidence for early diagenetic incorporation of biomarker lipids by covalent binding into benthic mat sediments from a salt pond Guerrero Negro, Mexico. They used sequential chemolysis and HyPy degradation on extracted microbial mat sediments and found evidence for early diagenetic incorporation of a variety of linear, branched and polycyclic lipid skeletons into proto-kerogen on a timescale of only years to decades. The lipids includes bound hopanoids and bound steroids.

So, this supports the idea that HyPy is an effective method for trying to detect bound steroids in mat proto-kerogen due to the I) high sample capacity and ii) use of reducing conditions that yields appreciable steranes and sterene products. It further supports the idea as described in 2) that the Py-GC-MS method used in this investigation is maybe problematic for detecting immature bound steroids from proto-kerogen.

It is surprising since this group has their own HyPy equipment that they choose an online Py-GC-MS method to try and detect kerogen-bound steroids.

The amount of high mw and polar material in pyrolysates from "young" mat sediments will be appreciable so it is best to choose a method that generates a substantial portion of bound steroids as hydrocarbon products. Even with HyPy, the "polar" fraction dominates the pyrolysate products so this problem will be even more acute with Py-GC-MS performed with an inert gas (rather than high pressure hydrogen and a catalyst as used in HyPy)."

**REPLY**: We acknowledge the article recently published by Lee et al. (2019) and have included it in our revised manuscript. However, in contrast to our results, the concentrations of steranes and hopanes in their study are in the same order. On the other hand, the HyPy equipment in our group was broken during the time of this project. Nevertheless, we feel that the proven suitability of HyPy does not reject the applicability of Py-GC-MS for this study. As also indicated in the response to referee #1, several studies have demonstrated that Py-GC-MS is a suitable method to investigate steroids in immature kerogen (Gelin et al., 1996; Kruge and Permanyer, 2004).

**Changes in manuscript**: We have discussed the suggested paper (Lee et al., 2019) in the revised version (chapter 4.2.3). The applicability of Py-GC-MS and the differences compared to HyPy have been also indicated in chapter 3.5 and 4.2.3.

References cited in the reply:

Bühring S. I., Smittenberg R. H., Sachse D., Lipp J. S., Golubic S., Sachs J. P., Hinrichs K. U. and Summons R. E. (2009). A hypersaline microbial mat from the Pacific Atoll Kiritimati: insights into composition and carbon fixation using biomarker analyses and a $^{13}$C-labeling approach. *Geobiology* 7, 308–323.

Gelin F., Sinninghe Damsté J. S., Harrison W. N., Reiss C., Maxwell J. R. and De Leeuw J. W. (1996). Variations in origin and composition of kerogen constituents as revealed by analytical pyrolysis of immature kerogens before and after desulphurization. *Organic Geochemistry* 24, 705–714.

Kruge M. A. and Permanyer A. (2004). Application of pyrolysis-GC/MS for rapid assessment of organic contamination in sediments from Barcelona harbor. *Organic Geochemistry* 35, 1395–1408.

[revised manuscript text omitted]

---

## Author Response (AR2)

**Responses from the authors to the comments of the subject editor (Dr. Marcel van der Meer) on manuscript bg-2019-124**

We sincerely thank Dr. van der Meer for the constructive and thoughtful comments that helped to improve our manuscript.

*General comments: It is a pleasure for me to accept your manuscript for publication in Biogeosciences. I do have a few minor issues, in the track changes version of you manuscript with the changes in red.*

*Specific points:*

1.  In line 19 and 24 on page 1 you use as "low in abundance as compared to" and "was low as compared to". I think you can lose the "as". They were low compared to something else.

    **Rephrased accordingly. See track changes Page 1, Line 19 and 24.**

2.  Page 3, line 2: a barrier against the preservation of eukaryotic lipids? Not sure what you meant here. I am guessing that all lipids are turned over, but since there are no Eukaryots in deeper layers their lipids are only found in the top layer. Heterotrophic bacteria, their lipids are probably also turned over, but since heterotrophic bacteria are still present at deeper layers you still find that type of lipids. I am guessing there is a chemical barrier for the presence of Eukaryots only in the top layer, oxygen perhaps?

    **Changed to "a significant mechanical and chemical barrier against the preservation of lipids sourced from water column and upper mat layers". See track changes Page 3, Line 2.**

3.  Page 3, Line 5: primary ecological signal? primary production signal? I am not sure what you mean by primary ecological signal.

    **Changed to "primary production signal". See track changes Page 3, Line 5.**

4.  Page 3, Line 24 (here refers to kerogen fraction).

    **Deleted "(here refers to kerogen fraction)". See track changes Page 3, Line 24.**

5.  Page 4, line 12: heavy evaporation, strong evaporation?

    **Changed to "strong evaporation". See track changes Page 4, Line 12.**

6.  Page 4, Line 18: ancient mat generations? older compacted mat generations (unless you know they are ancient?).

    **Rephrased to "older mat generations". See track changes Page 4, Line 18.**

7. Page 4, Line 19: You mean these older more compacted mat remnants are still degraded today. Recent anaerobic microorganisms, I think you can lose the recent.

    **Changed to "active anaerobic microorganisms". See track changes Page 4, Line 19**

8. Page 5, line 5: hopanoic acids? At least in the fatty acid fraction (same in line 17).

    **Changed to "hopanoic acids". See track changes Page 5, Line 5 and 17.**

9. Page 6, line 31: the earlier, buried, growth phases showed... or the earlier, deeper, growth phases....

    **Rephrased accordingly to "The earlier, deeper growth phase showed…". See track changes Page 6, Line 29.**

10. Page 10, line 12: 10 to the 2nd is an order more abundant than 10 to 1st.

    **Modified to "where carbonate-bound FAs were an order less abundant than free lipids (Table S3 and S4)". See track changes Page 10, Line 12.**

11. Page 11, line 12: "either algae or terrestrial plant, more over the compounds are known to be produced by some algae"... That is what you just said, produced by algae?

    **Modified to "…, potentially indicating contributions from either algae including diatoms (Rampen et al., 2010; Volkman, 2003) or terrestrial plants (Volkman, 1986)". See track changes Page 11, Line 12.**

12. Page 11, Line 21: recycling of organic matter back to CO2 could also lead to 13C depleted CO2 and therefore 13C depleted primary productivity.

    **It is our opinion that the provided potential reasons in the manuscript should be sufficient to explain the enriched $^{13}C$ values of individual lipids in our studied mat, and we see no reason to provide complex theories to explain the commonly observed $^{13}C$ values reported in the other studies.**

13. Page 12, line 21: "an outstanding behavior" I would say "a different rend with depth".

    **Changed to "a different trend with depth". See track changes Page 12, Line 23.**

[revised manuscript text omitted]